# GATA transcription factors, SOX17 and TFAP2C, drive the human germ-cell specification program

Yoji Kojima[1,2,3] , Chika Yamashiro[1,2] , Yusuke Murase[1,2], Yukihiro Yabuta[1,2], Ikuhiro Okamoto[1,2], Chizuru Iwatani[4], Hideaki Tsuchiya[4], Masataka Nakaya[1,4], Tomoyuki Tsukiyama[1,4] , Tomonori Nakamura[1,2,7] , Takuya Yamamoto[1,3,5,6], Mitinori Saitou[1,2,3]

The in vitro reconstitution of human germ-cell development provides a robust framework for clarifying key underlying mechanisms. Here, we explored transcription factors (TFs) that engender the germ-cell fate in their pluripotent precursors. Unexpectedly, *SOX17*, *TFAP2C*, and *BLIMP1*, which act under the BMP signaling and are indispensable for human primordial germ-cell-like cell (hPGCLC) specification, failed to induce hPGCLCs. In contrast, *GATA3* or *GATA2*, immediate BMP effectors, combined with *SOX17* and *TFAP2C*, generated hPGCLCs. *GATA3/GATA2* knockouts dose-dependently impaired BMP-induced hPGCLC specification, whereas *GATA3/GATA2* expression remained unaffected in *SOX17*, *TFAP2C*, or *BLIMP1* knockouts. In cynomolgus monkeys, a key model for human development, *GATA3*, *SOX17*, and *TFAP2C* were co-expressed exclusively in early PGCs. Crucially, the TF-induced hPGCLCs acquired a hallmark of bona fide hPGCs to undergo epigenetic reprogramming and mature into oogonia/gonocytes in xenogeneic reconstituted ovaries. By uncovering a TF circuitry driving the germ line program, our study provides a paradigm for TF-based human gametogenesis.

## Introduction

Germ cells are the carriers of genetic as well as epigenetic information into new individuals, and thus serve as an enduring link between generations. Accordingly, they bear a capacity to replicate genetic information with high fidelity (1, 2, 3). On the other hand, they also create genetic and epigenetic diversity through meiotic recombination and epigenetic reprogramming/programming, respectively, providing a driving force for evolution (4, 5). Anomalies in such processes often lead to diseased states, including infertility and genetic/epigenetic disorders of offspring. Therefore, investigations into the mechanism of germ-cell development not only promote our understanding of fundamental principles of heredity and evolution but also provide salient information regarding the etiology of critical diseases.

Despite such importance, investigations into human germ-cell development have been limited because of the difficulties in accessing relevant experimental materials and ethical restrictions. Notably, recent advancements in the in vitro reconstitution of human germ-cell development using human pluripotent stem cells (hPSCs), including embryonic stem cells (hESCs) and induced pluripotent stem cells (hiPSCs), have created novel opportunities for such studies, permitting investigations into the mechanisms of human germ-cell development as an emerging frontier in reproductive biology/medicine (6, 7). Accordingly, hPSCs are induced into cells bearing properties similar to human primordial germ cells (hPGCs) (8, 9), the founding population of the human germ-cell lineage that eventually gives rise to either spermatozoa or oocytes. The induced hPGC-like cells (hPGCLCs) are further differentiated into oogonia/early oocyte-like cells with appropriate epigenetic reprogramming in a reconstituted ovary culture (10, 11), or into pro-spermatogonia-like cells in a reconstituted testis culture (12). Although further reconstitution of human germ-cell development remains a key challenge, these advances recapitulate a period of more than 10 wk of human germ-cell development, leading to a number of key findings with regard to the mechanism of human germ-cell development in general, and germ-cell specification in particular (8, 9, 13, 14, 15, 16, 17).

In the case of germ-cell specification, humans as well as non-human primates such as cynomolgus monkeys (*Macaca fascicularis*), use transcriptional and signaling programs evolutionarily distinct from those in mice, which have long been a paradigm for mammalian development (8, 9, 13, 18, 19). Specifically, in humans, WNT signaling induces *EOMES*, which, together with bone

[1]Institute for the Advanced Study of Human Biology (ASHBi), Kyoto University, Yoshida-Konoe-cho, Kyoto, Japan    [2]Department of Anatomy and Cell Biology, Graduate School of Medicine, Kyoto University, Yoshida-Konoe-cho, Kyoto, Japan    [3]Center for iPS Cell Research and Application (CiRA), Kyoto University, Shogoin-Kawahara-cho, Kyoto, Japan    [4]Research Center for Animal Life Science, Shiga University of Medical Science, Seta-Tsukinowa-cho, Otsu, Japan    [5]AMED-CREST, AMED, Tokyo, Japan    [6]Medical-Risk Avoidance Based on iPS Cells Team, RIKEN Center for Advanced Intelligence Project (AIP), Kyoto, Japan    [7]The Hakubi Center for Advanced Research, Kyoto University, Yoshida-Konoe-cho, Kyoto, Japan

Correspondence: ykojima@cira.kyoto-u.ac.jp; saitou@anat2.med.kyoto-u.ac.jp

morphogenetic protein 4 (BMP4) signaling, induces *SOX17* as one of the most upstream transcription factors (TFs) for hPGC(LC) specification (13). *SOX17* is essential for the expression of key downstream genes, including *BLIMP1,* and for activating other germ-cell specification programs (8, 13). TFAP2C also serves as a key upstream TF that functions in parallel and in an interdependent fashion with SOX17 and is critical for the repression of somatic programs (13, 15). Such programs for germ-cell specification appear to be relatively well conserved in cynomolgus monkeys (18, 19). In contrast, in mice, *Sox17* has no role in germ-cell specification (20), and BMP4 signaling activates endogenous WNT signaling that in turn induces *T* (*T* has no role in humans (13)), which up-regulates *Blimp1* and *Prdm14*, two of the most upstream TFs for germ-cell specification (21, 22, 23). *Blimp1*, *Prdm14*, and *Tfap2c* are essential and sufficient for the global control of downstream programs, including by reactivating pluripotency programs, repressing somatic programs, and initiating epigenetic reprogramming (22, 23, 24, 25). These findings demonstrate that the TFs and TF hierarchies involved in conferring the germ-cell fate in humans are distinct from those in mice, highlighting the importance of further promoting human germ-cell biology.

In regard to the mechanism of human germ-cell specification, a fundamental question remains to be answered: That is, which TFs or TF combinations are sufficient to give rise to the germ-cell fate in their precursors? The answer to this question could help establish a foundation for TF-based human gametogenesis. In mice, three TFs (*Blimp1*, *Prdm14*, and *Tfac2c*), and to a lesser extent, two TFs (*Blimp1* and *Tfap2c*; *Prdm14* and *Tfap2c*) or a single TF (*Prdm14*), are sufficient to confer the germ-cell fate to their precursors, and such TF-induced mouse PGCLCs (mPGCLCs) contribute to spermatogenesis (25). We therefore set out to define the TFs that replace the BMP4 signaling and are sufficient to establish the identity of hPGCs on their precursors. Unexpectedly, we found that three TFs that are essential for hPGCLC specification—that is, *SOX17*, *TFAP2C*, and *BLIMP1*—are nonetheless not sufficient, and in contrast, the GATA family of TFs, combined with *SOX17* and *TFAP2C*, drives the hPGCLC program.

## Results

### *SOX17*, *TFAP2C*, and *BLIMP1* are not sufficient to generate hPGCLCs

For hPGCLC induction, hiPSCs are first induced into incipient mesoderm-like cells (iMeLCs) by stimulating with activin A and a WNT signal activator (CHIR99021) for 2 d, and iMeLCs are then induced into hPGCLCs by stimulating with bone morphogenetic protein 4, together with proliferation/survival factors, including stem cell factor (SCF), EGF, and leukemia inhibitory factor (LIF), under a floating aggregate condition (9, 13, 26). hPGCLCs that express key genes such as *SOX17*, *TFAP2C*, *BLIMP1*, and *NANOS3* are induced as early as day 2 of induction (d2 hPGCLCs), show a progressive maturation, and persist at least until around d10 (9, 13, 26).

We set out to identify TFs that are sufficient to confer the germ-cell fate on iMeLCs in the absence of BMP signaling. At the outset, we evaluated whether *SOX17*, *TFAP2C*, or *BLIMP1*, three TFs essential

for hPGCLC specification (8, 9, 13), would be sufficient to induce the germ-cell fate when expressed either singly or in one of various combinations. For this purpose, hiPSCs bearing the *BLIMP1-2A-tdTomato* (BT) and *TFAP2C-2A-EGFP* (AG) alleles (585B1 BTAG (XY)) (9) were transfected with *piggyBac*-based vectors expressing (i) the reverse tetracycline trans-activator (rtTA) under a constitutively active promoter and (ii) the genes of interest (*SOX17*, *TFAP2C*, *SOX17/TFAP2C*, *SOX17/BLIMP1*, *TFAP2C/BLIMP1*, or *SOX17/TFAP2C/BLIMP1*) under the control of tetracycline regulatory elements with transcription termination by the rabbit *β*-globin poly A sequence (*rBGpA*), so that the genes of interest exhibited timed expression in a doxycycline (Dox)-dependent manner and could be distinguished from the endogenous ones by the presence of *rBGpA* (Fig 1A). For each transfectant, we selected two clones that exhibited transgene expression levels in hiPSCs comparable with the corresponding endogenous gene expression levels in hPGCLCs (Figs 1B and S1A). The expression of the transgenes in a clone expressing *SOX17/TFAP2C/BLIMP1* was confirmed with Western blotting (Fig S1B). All the hiPSC clones selected exhibited undifferentiated morphology.

We first examined the effects of the transgene expression in hiPSCs cultured with Dox for 24 h. Quantitative Real Time-PCR (qRT-PCR) for the endogenous key loci (*SOX17*, *TFAP2C*, *BLIMP1*, and *NANOS3*) showed that no clones up-regulated *SOX17*, whereas the *SOX17*, *SOX17/TFAP2C*, and *SOX17/TFAP2C/BLIMP1* clones up-regulated *TFAP2C* mildly and *BLIMP1* to an extent comparable with that in d2 hPGCLCs (Fig 1B). The *TFAP2C/BLIMP1*, *SOX17/TFAP2C*, and *SOX17/TFAP2C/BLIMP1* clones up-regulated endogenous *NANOS3* to a level similar to that in d2 hPGCLCs (Fig 1B). Because the *TFAP2C* clones had no impact on all these genes (*TFAP2C* appeared to repress endogenous *TFAP2C*) (Fig 1B), we excluded them from the subsequent analyses.

We next analyzed whether the expression of these genes in iMeLCs might induce the germ-cell fate (Fig 1C). The iMeLCs induced by activin A and CHIR99021 from all the clones bore a morphology indistinguishable from that of the parental clone (Fig 1D). Upon d4 of induction by BMP4 or BMP4 and Dox, iMeLC aggregates from all the clones exhibited a distinct cluster of BT-positive (BT$^+$) and AG-positive (AG$^+$) cells, as revealed by observation under a fluorescence dissection microscope or FACS (Fig 1E). We noted that the *SOX17/TFAP2C* clones stimulated by BMP4 and Dox, although forming small aggregates, differentiated into BT$^+$AG$^+$ cells at a very high efficiency (~90%), whereas the other clones formed BT$^+$AG$^+$ cells with an efficiency of ~20–30% (Fig 1E and F). This might be because *SOX17* and *TFAP2C* expression could be a rate-limiting event for hPGCLC specification, and the Dox-induced expression of *SOX17* and *TFAP2C* would create a state highly competent for BMP-induced hPGCLC specification. In addition, the iMeLC aggregates of the *SOX17*, *SOX17/TFAP2C*, and *TFAP2C/BLIMP1* clones became smaller when stimulated with BMP4 and Dox, which might have been due to a subtle but significant difference in the expression levels of *SOX17*, *TFAP2C*, or *BLIMP1* (e.g., *Blimp1*/*BLIMP1* is known to induce cell-cycle arrest in various contexts (27, 28)).

In contrast, with Dox stimulation alone, no iMeLC aggregates showed BT$^+$AG$^+$ cells (Fig 1E and F). Upon stimulation with Dox, the *SOX17* clones induced weak BT$^+$ cells, the *SOX17/BLIMP1* clones showed no BTAG positivity, the *TFAP2C/BLIMP1* clones generated small aggregates with weak BTAG positivity (less than the threshold

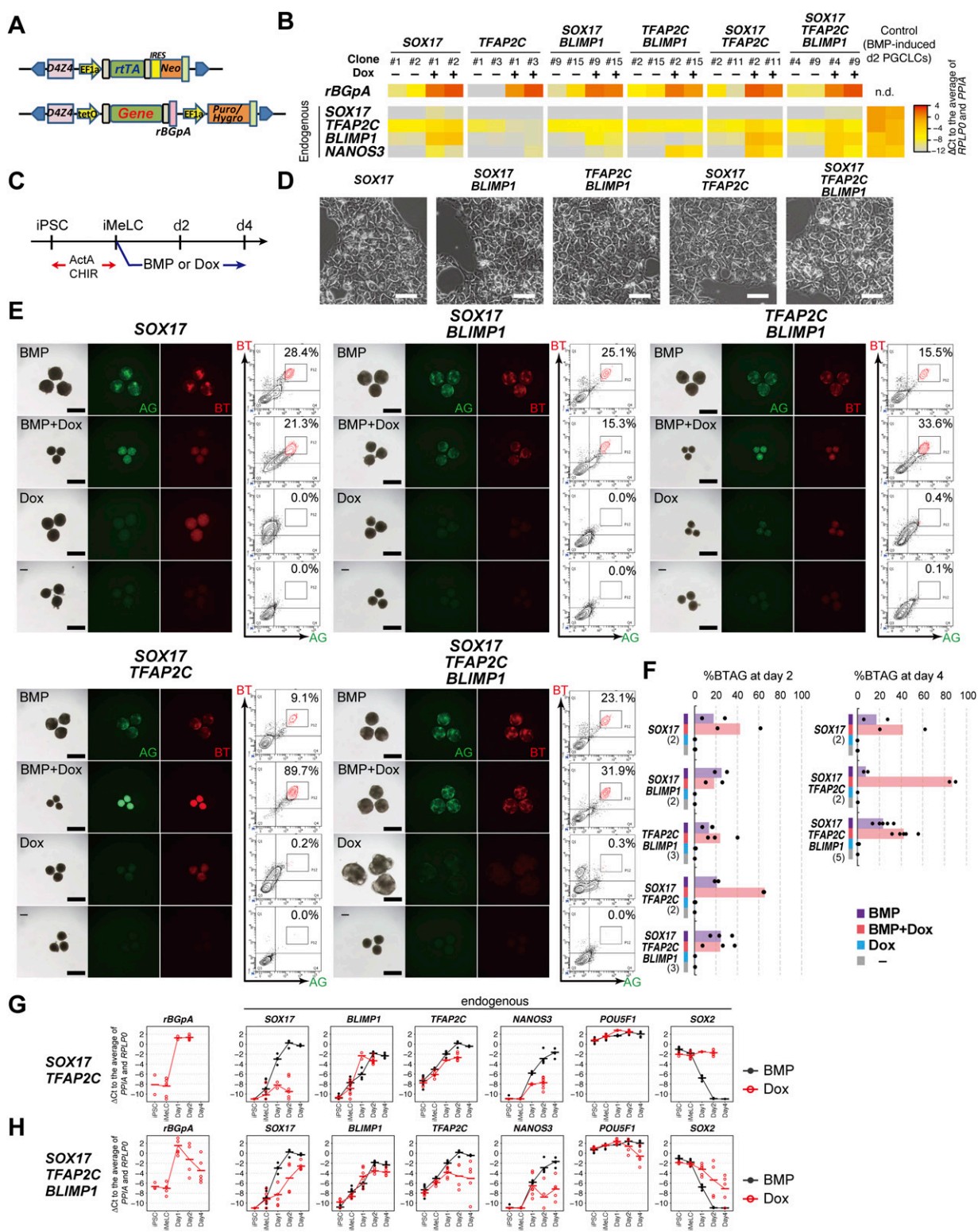

**Figure 1.** *SOX17*, *TFAP2C*, and *BLIMP1* do not create human primordial germ-cell-like cells (hPGCLCs).

**(A)** A schematic of the vectors used for Dox-inducible expression. The coding sequence of each gene was cloned in the designated position. D4Z4: D4Z4 macrosatellite repeat insulator; EF1α: promoter sequence of human *EEF1A1*; rtTA: reverse tetracycline trans-activator; IRES: internal ribosome entry site; Neo: Neomycin resistance gene; tetO: Tet operator sequence; rBGpA: rabbit β-globin polyadenylation signal; Puro/Hygro: resistance gene for puromycin/hygromycin. **(B)** Heat map representation of the expression levels of the indicated genes in the designated hiPSC clones stimulated with (+) or without (−) Dox (1.0 μg/ml) for 24 h. Two clones were examined for each transgene combination. To quantify the expression levels of the transgenes or endogenous genes by qRT-PCR, primer pairs for the *rBGpA* or 3′ untranslated regions were

levels), the *SOX17/TFAP2C* clones induced weak BT⁺ cells, and the *SOX17/TFAP2C/BLIMP1* clones induced weak BT⁺ or AG⁺ cells, with the generation of very minor BT⁺AG⁺ cells (<1%) (Fig 1E and F). Because very few/no BT⁺AG⁺ cells were induced under any conditions, we characterized the properties of BT⁺ cells induced by the transgenes by qRT-PCR. All such cells failed to show a proper up-regulation of endogenous *SOX17*, *TFAP2C*, and *NANOS3* and down-regulation of *SOX2*, despite their substantial transgene expression (Figs 1G and H and S1C and D).

Taken together, these facts lead us to conclude that no combination of *SOX17*, *TFAP2C*, and *BLIMP1* is sufficient to induce appropriate endogenous *SOX17/TFAP2C* expression and other hPGCLC properties, including *NANOS3* expression and *SOX2* repression, in iMeLCs. Thus, the three TFs are not sufficient to replace the BMP4 signaling for hPGCLC induction. On the other hand, the finding that all the clones generated BT⁺AG⁺ cells upon BMP4 and Dox provision indicates that the expression of *SOX17/TFAP2C/BLIMP1* transgenes does not interfere with the hPGC specification program.

## Exploration of relevant TFs by transcriptome analysis

To explore the effects of the TF expression more globally and to identify a relevant TF(s) that might be required for hPGCLC generation, we compared the transcriptome dynamics associated with the TF expression to those of the parental clone induced by BMP4 by RNA-sequencing (RNA-seq) (13, 29) (see Table S1 for the RNA-seq samples analyzed in this study).

In agreement with the previous report (13), principal component analysis (PCA) revealed that parental cells exhibited characteristic cell–property transitions during hPGCLC induction: on the PC1/2 plane, they transitioned along a V-shaped trajectory, with hiPSCs and iMeLCs plotted on the top-left corner, d1 iMeLC aggregates progressing diagonally toward the bottom, and d2/d4 BT⁺AG⁺ cells progressing up toward the top-right corner (Fig 2A). In contrast, whereas hiPSCs/iMeLCs of all the TF-expressing clones (no Dox provision) were plotted at positions similar to those of parental hiPSCs/iMeLCs, upon Dox provision on iMeLC aggregates (i.e., TF expression), all the clones shifted their positions in parallel along the PC1 coordinate with a retention or some down-regulation of their PC2 scores, but none of the clones acquired a state similar to d4 hPGCLCs (Fig 2B–E). We examined the expression of 481 genes that characterize the hPGCLC specification process (13), which revealed that the d2 TF-expressing cells failed to show a property similar to those of d2/d4 BT⁺AG⁺ cells (Fig S2A). Most notably, they

lacked sufficient expression of the genes specifying hPGCLC properties (Fig S2A).

With a focus on the *SOX17/TFAP2C/BLIMP1* clones, we determined the differentially expressed genes (DEGs) between BMP4- and Dox-stimulated cells at d1 (whole iMeLC aggregates) and d2 (BMP4: BT⁺AG⁺ cells; Dox: BT⁺ cells) of the respective stimulations. Compared with the Dox-stimulated cells, the BMP4-stimulated cells up-regulated 136 and 26 genes and down-regulated 104 and 73 genes at d1 and d2, respectively (Fig 2F and Table S2) (note that the numbers of DEGs were smaller at d2 because of a variability in gene expression of the BT⁺ cells of the Dox-induced *SOX17/TFAP2C/BLIMP1* clones [Fig 2B]). The genes up-regulated in BMP4-stimulated cells at d1 and/or d2 were enriched with those for "transcription from RNA polymerase II promoter" (Gene Ontology [GO] functional terms), "signaling pathways regulating pluripotency of stem cells," and "WNT/HIPPO/TGF-*β* signaling pathways" (Kyoto Encyclopedia of Genes and Genomes [KEGG] pathway) (Fig 2G and Table S2), and included known BMP targets such as *GATA3*, *TFAP2A*, *MSX1*, *EVX1*, *HAND1*, *TBX3*, *MSX2*, and *CDX2* (Figs 2G and H and S2B and Table S2) (Note that *TFAP2A*, *HAND1*, *HAPLN1*, *MSX2*, and *CDX2* were highly up-regulated at both d1 [Fig 2H] and d2 [Fig S2B]). The genes up-regulated in Dox-stimulated cells at d1 and/or d2 were enriched with those for "cellular response to glucose stimulus" (GO terms) and "PI3K-AKT signaling pathway" (KEGG pathways) (Fig 2I and Table S2), and included *H19*, *DLL1*, *GSTA1*, *COL14A1*, *TCF7L1*, *HHEX*, *NRP1*, *CHODL*, and *LEFTY1* (Figs 2I and S2C and Table S2), some of which are characteristic for anterior epiblast/endoderm in gastrulating mouse embryos (32, 33, 34). The DEGs between the *SOX17/TFAP2C* clone-derived BMP4/Dox- and Dox-stimulated cells at d2 (BMP4/Dox: BT⁺AG⁺ cells; Dox: BT⁺ cells) included similar genes (Fig S2D and E and Table S2). These findings raised a possibility that, unlike in mice (25), a TF(s) recognized as a canonical BMP target(s) may play a key role in conferring the germ-cell fate on iMeLCs.

To explore the relevance of such TFs in an in vivo context, we examined their expression in the single-cell transcriptome of PGCs (cyPGCs) of cynomolgus monkeys, a primate model for human development (18, 30). Among the TFs examined, *GATA3* and *MSX2* were expressed at relatively high levels in cyPGCs from embryonic day (E) 13 to E17 (early cyPGCs) as well as in cy germ cells from E36 to E51 (oogonia/gonocytes), and whereas *TBX3* and *HAND1* were expressed in early cyPGCs (at lower levels than *GATA3* and *MSX2*), their expression was sporadic/repressed in oogonia/gonocytes (Fig 2J). The other TFs showed more sporadic/no expression in early cy germ cells (Fig 2J). Accordingly, we decided to focus on exploring the

used, respectively, and the ΔCt was calculated from the average Ct value of two housekeeping genes, *RPLP0* and *PPIA* (set as 0). **(C)** The protocol for hPGCLC induction. iMeLC aggregates were induced for hPGCLC fate by bone morphogenetic protein 4 (BMP4) or Dox (1.0 *μ*g/ml) in the presence of stem cell factor, EGF, and leukemia inhibitory factor. ActA: activin A; CHIR: CHIR99021. **(D)** Phase-contrast images of iMeLCs in the designated clones. No apparent morphological differences were seen among the clones. Representative images of at least two independent experiments are shown (shown in Fig 1F). Bar: 50 *μ*m. **(E)** Bright-field and fluorescence (*TFAP2C-EGFP* [AG] and *BLIMP1-tdTomato* [BT]) images, and FACS analyses for BTAG expression in floating aggregates of the indicated transgene-expressing clones at day 4 of the indicated stimulation. (–): induction only with stem cell factor, EGF and leukemia inhibitory factor. Representative images of at least two independent experiments are shown (see Fig 1F). Bars, 200 *μ*m. **(F)** Percentage of BT⁺AG⁺ cells of the indicated transgene-expressing clones with the indicated stimulations at day 2 (left) and day 4 (right). Dots represent values for each experiment and the bars represent their averages. The numbers of inductions performed are shown in parenthesis. **(G, H)** Expression dynamics of *rBGpA* (transgenes) and the indicated endogenous genes in the *SOX17/TFAP2C* (G) and *SOX17/TFAP2C/BLIMP1* (H) clones induced by BMP4 (black) or Dox (red). d1: whole aggregates; d2/d4: BT⁺AG⁺ cells for induction by BMP4, BT⁺ cells for induction by Dox. For each gene, the ΔCt was calculated from the average Ct value of two housekeeping genes, *RPLP0* and *PPIA* (set as 0). **(G, H)** Three independent experiments with two *SOX17/TFAP2C* clones (G) and three *SOX17/TFAP2C/BLIMP1* clones (H) were performed.

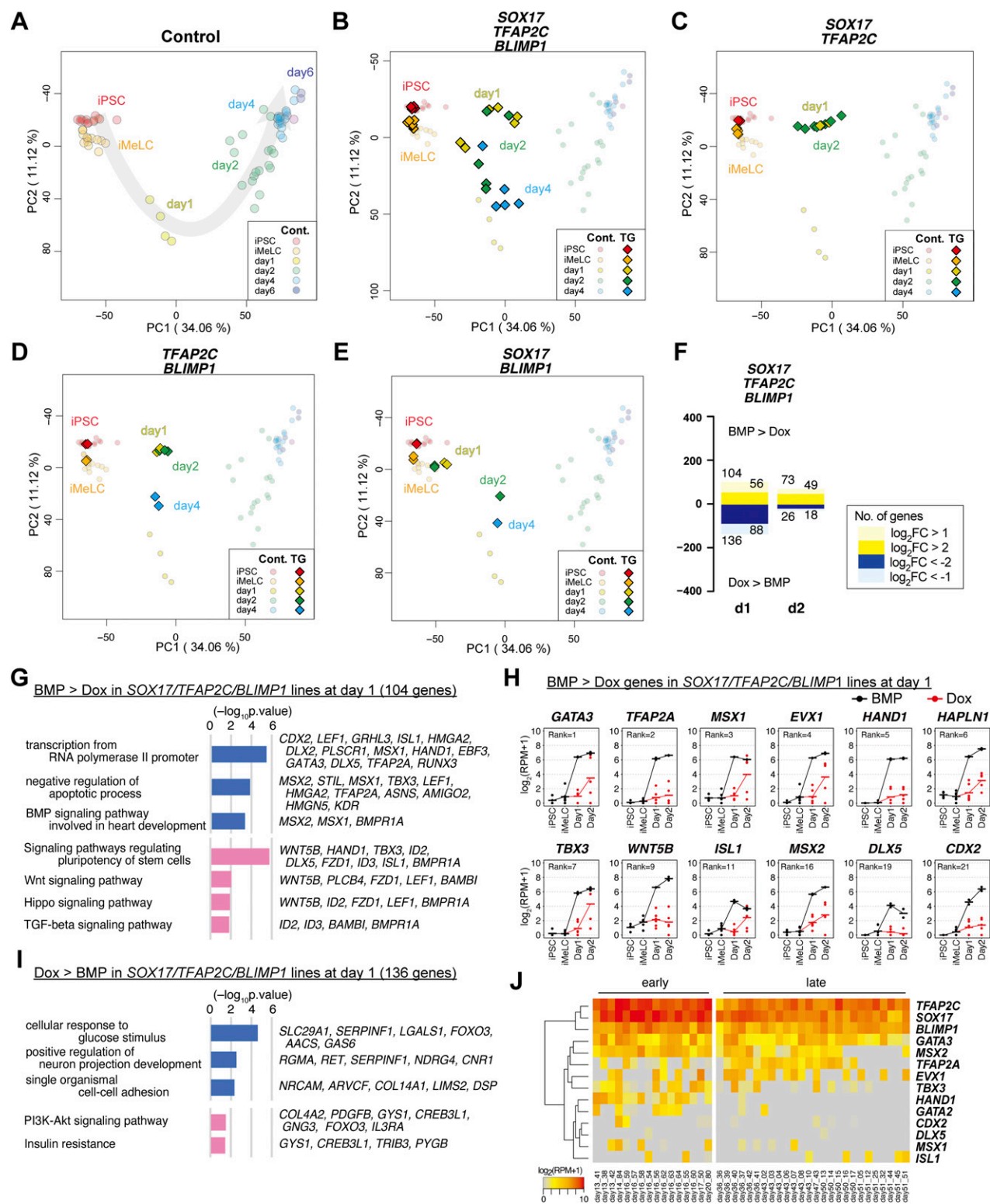

**Figure 2. Transcriptome analysis of the effects of the transcription factor expression.**
(A, B, C, D, E) Principal component analysis (PCA) of the effects of the transcription factor expression (see the Materials and Methods section for details). **(A)** The PCA plots of the cells (hiPSCs, iMeLCs, d1 whole aggregates, d2/d4/d6 BT⁺AG⁺ cells) derived from the parental clone (585B1 BTAG). The developmental progression is indicated by an arrow. **(B, C, D, E)** The PCA plots of the *SOX17/TFAP2C/BLIMP1* (B), *SOX17/TFAP2C* (C), *TFAP2C/BLIMP1* (D), and *SOX17/BLIMP1* (E) clones (squares) are overlaid with those of the parental clone (circles with pale color). See Table S1 for the samples analyzed. The color coding is as indicated. **(F)** The numbers of the differentially expressed genes at d1/d2 between bone morphogenetic protein (BMP)– and Dox-stimulated cells of the *SOX17/TFAP2C/BLIMP1* clones (*P* < 0.01 by Tukey–Kramer test, log₂[RPM + 1] >

function of *GATA3* and *MSX2*. They were indeed expressed at high levels upon induction of hPGCLCs by BMP4 (Fig 3A), and in the iMeLC aggregates of the *SOX17*/*TFAP2C* and *SOX17*/*TFAP2C*/*BLIMP1* clones stimulated by BMP4 and Dox (Fig S3A).

### GATA TFs, *SOX17*, and *TFAP2C* drive hPGCLC induction

*GATA3* is a member of the evolutionarily conserved GATA family of TFs, which bears six paralogs in vertebrates and plays key roles in the specification of a wide range of cell types in three germ layers as well as in extraembryonic tissues (see reference 35 for review; see the Discussion section for the roles of *GATA3* in relevant contexts), but its role in germ-cell development has not been reported. *MSX2* is a member of the evolutionarily conserved homeodomain TFs, which bear three and two paralogs in mice and humans, respectively, and play important functions during neural tube, tooth, and limb development (see reference 36 for review). In mice, *Msx1* and *Msx2* double mutants show a defect in meiotic prophase in female embryonic germ cells (37).

We decided to examine the effect of *GATA3* or *MSX2* expression combined with the expression of *SOX17* and *TFAP2C*, for the following reasons: (1) both *SOX17* and *TFAP2C* are required for hPGCLC specification (8, 13); (2) *SOX17* and *TFAP2C* did not activate each other (Fig 1B and E–H); (3) *BLIMP1* was activated by *SOX17* (Fig 1B and E) (13); (4) nearly all the cells in the *SOX17*/*TFAP2C* clone-derived iMeLC aggregates became BT⁺AG⁺ in response to BMP4 and Dox (Fig 1E and F). We isolated two independent clones expressing *MSX2*, *SOX17*, and *TFAP2C*, verified the transgene expression in hiPSCs by qRT-PCR (Fig S3B and C), and examined whether their expression in iMeLC aggregates (Fig S3D) induces the germ-cell fate; however, we found that *MSX2*, *SOX17*, and *TFAP2C* expression failed to induce BT⁺AG⁺ cells (Fig 3B).

Next, therefore, we isolated two independent clones expressing *GATA3*, *SOX17*, and *TFAP2C*, and verified the transgene expression in hiPSCs by qRT-PCR (Fig S3B). We found that *GATA3*, *SOX17*, and *TFAP2C* expression up-regulated endogenous *SOX17* to a moderate extent and endogenous *TFAP2C* to a substantial extent (Fig S3C). We then expressed these genes in iMeLC aggregates; remarkably, we observed a progressive induction of distinct populations of BT⁺AG⁺ cells by d4 of the transgene induction (Figs 3C and S3D and E). We then isolated induced BT⁺AG⁺ cells at d4, and examined their expression of key genes by qRT-PCR. As shown in Fig 3D, they up-regulated endogenous *SOX17*, *TFAP2C*, *BLIMP1*, and *NANOS3* to levels indistinguishable from those in BT⁺AG⁺ cells induced by BMP4, and repressed *SOX2* to a great extent.

We next examined whether other GATA TFs might also induce the BT⁺AG⁺ cells in iMeLCs. We decided to explore the function of *GATA2* because (1) *GATA2* is also up-regulated upon hPGCLC induction,

although at a lower level than *GATA3* (Fig 3A), and is expressed in early cyPGCs (*GATA2* is detectable in at least 6 of 16 early cy germ cells) (Fig 2J); (2) GATA2 shows the highest structural similarity to GATA3 among the GATA TFs (35, 38); and (3) GATA3 and GATA2 show a compensatory function and bear overlapping genome-wide binding profiles in other relevant contexts (35, 39, 40, 41). Accordingly, we isolated a clone expressing *GATA2*, *SOX17*, and *TFAP2C*, verified the transgene expression in hiPSCs, and found that *GATA2*, *SOX17*, and *TFAP2C* expression up-regulated both endogenous *SOX17* and *TFAP2C* to a substantial extent (Fig S3C). Consistent with this result, the expression of *GATA2*, *SOX17*, and *TFAP2C* in iMeLCs robustly induced BT⁺AG⁺ cells with an expression profile of key genes similar to that in BT⁺AG⁺ cells induced by *GATA3*, *SOX17*, and *TFAP2C* or BMP4 (Fig 3C and D). Interestingly, however, the sizes/cell numbers of the iMeLC aggregates induced with *GATA2*, *SOX17*, and *TFAP2C* were smaller than those induced with *GATA3*, *SOX17*, and *TFAP2C* or BMP4 (Figs 3C and S3F), suggesting that *GATA3* and *GATA2* play overlapping but distinct functions in iMeLC aggregates.

A recent report has shown that in hESCs, BMP signaling activates *GATA3*, which in turn up-regulates *BMP4*, thereby creating a feed-forward loop for persistent activation of the BMP signaling pathway (42). To exclude the possibility that *GATA3*/*GATA2*, *SOX17*, and *TFAP2C* expression activates the BMP signaling that in turn induces the BT⁺AG⁺ cells, we induced *GATA3*/*GATA2*, *SOX17*, and *TFAP2C* in iMeLCs in the presence of LDN193189, a potent inhibitor of the key receptor for BMP signaling, ALK2/3 (43, 44). As shown in Fig 3E and F, whereas the provision of LDN193189 blocked the induction of BT⁺AG⁺ cells by BMP4, it had little, if any, effect on the BT⁺AG⁺-cell induction by *GATA3*/*GATA2*, *SOX17*, and *TFAP2C*. Next, we explored the possibility that if *GATA3* up-regulates *BMP4* to a substantial extent, then the expression of *GATA3* with *SOX17*, but without *TFAP2C*, may also induce the BT⁺AG⁺ cells. For this purpose, we isolated a number of clones expressing *GATA3* and *SOX17*, verified the transgene expression (Fig S3B and C), and induced them in iMeLC aggregates (Fig S3D). Although the *GATA3*/*SOX17*-derived iMeLC aggregates up-regulated BT and activated AG to some extent, they did not form a distinct population of BT⁺AG⁺ cells (Fig S3E). Thus, *GATA3/2*, *SOX17*, and *TFAP2C* directly and cell-autonomously drive the hPGCLC program.

We determined the transcriptomes of the *GATA3*/*SOX17*/*TFAP2C* and *GATA3*/*SOX17* clones during the induction of BT⁺AG⁺/BT⁺ cells. PCA revealed that whereas the iMeLCs from the *GATA3*/*SOX17*/*TFAP2C* clones were nearly identical to wild-type iMeLCs, upon induction of the transgenes by Dox, they progressed directly toward the hPGCLC fate, bypassing the d1 iMeLC-aggregate state, and by d6 of induction, they acquired a transcriptome close to that of the d4/d6 hPGCLCs induced by BMP4 (Fig 4A). Accordingly, regarding the expression of 481 genes that characterize the hPGCLC specification

---

4 in cells with higher expression, log₂[fold change: FC] >1 [up, pale yellow; down, pale blue] or 2 [up, yellow; down, blue]). d1: iMeLC whole aggregates; d2: BT⁺AG⁺ and BT⁺ cells for BMP- and Dox-stimulated cells, respectively. Note that the numbers of differentially expressed genes were smaller at d2, because the gene expression of the BT⁺ cells of the Dox-induced *SOX17*/*TFAP2C*/*BLIMP1* clones was somewhat variable. **(G, I)** Gene ontology terms (blue) and KEGG pathways (pink) enriched in differentially expressed genes between BMP- and Dox-stimulated d1 *SOX17*/*TFAP2C*/*BLIMP1* clone aggregates. **(G, I)** Representative genes up-regulated in BMP- (G) or Dox- (I) stimulations and *P*-values are shown. **(F, H)** Expression dynamics of the genes up-regulated at d1 (F) in BMP-stimulated (black) compared with Dox-stimulated (red) *SOX17*/ *TFAP2C*/*BLIMP1* clone-derived cells. The ranks of the genes ordered by the fold changes between BMP and Dox stimulation are shown. Note that *TFAP2A*, *HAND1*, *HAPLN1*, *MSX2*, and *CDX2* were highly up-regulated in BMP-stimulated cells also at d2 (Fig S2B). See Table S1 for the samples analyzed. **(E, H, J)** Heat map representation of the expression of the genes in (H) in cynomolgus monkey fetal germ cells (early: embryonic day (E) 13-E17; late: E36-E51) (9, 18, 30, 31).

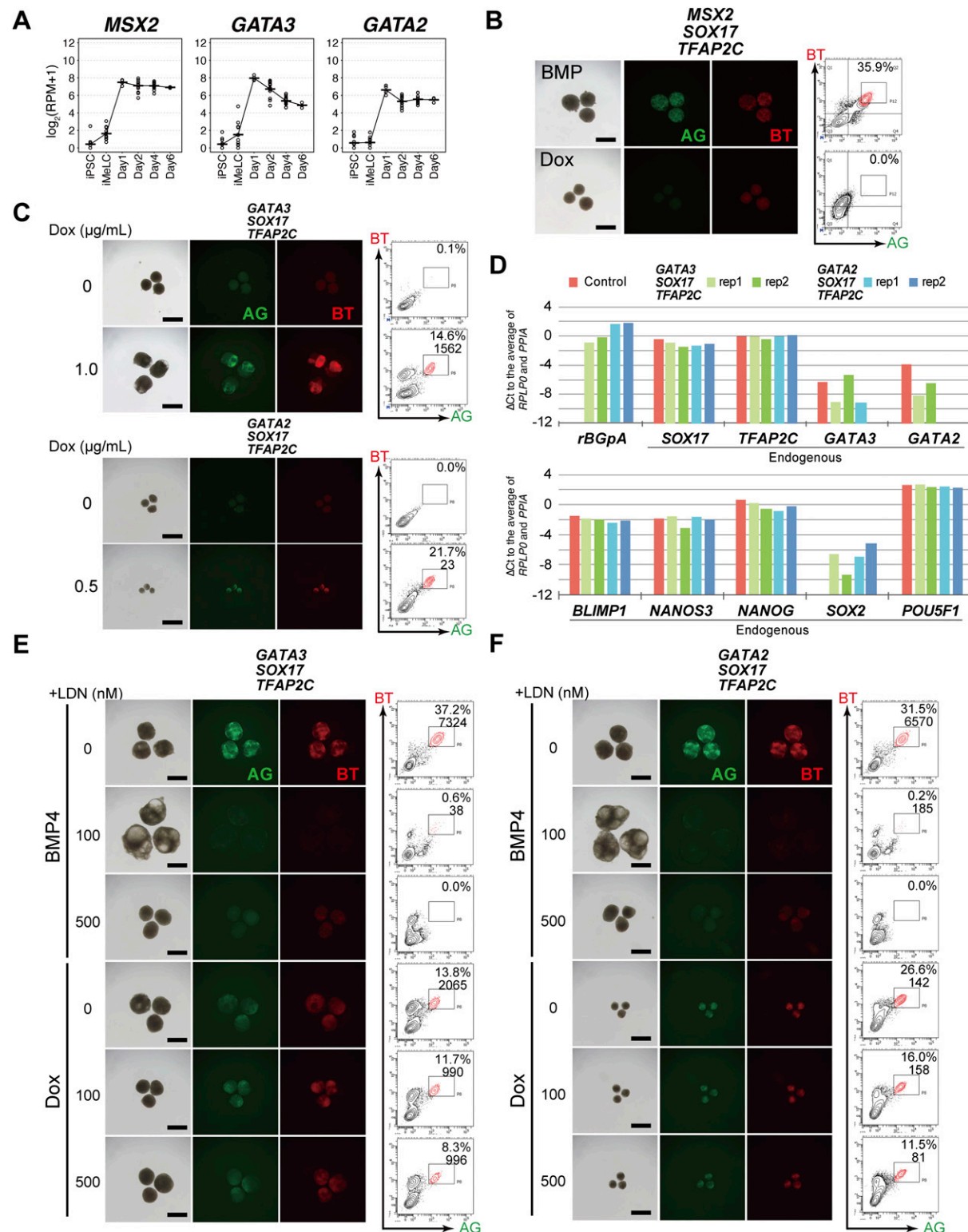

**Figure 3.   *GATA3/2*, *SOX17*, and *TFAP2C* generate human primordial germ-cell-like cells.**
**(A)** Expression dynamics of *MSX2*, *GATA3*, and *GATA2* during bone morphogenetic protein (BMP)-driven human primordial germ-cell–like cell induction from the parental hiPSCs. Log$_2$(RPM + 1) values are plotted and the bars indicate the mean value of each time point. See Table S1 for the samples analyzed. **(B)** Bright-field and fluorescence (AG/BT) images, and FACS analyses for BTAG expression in floating aggregates of the *MSX2*/*SOX17*/*TFAP2C* clone induced by BMP4 or Dox at d4. Representative images of three independent experiments are shown. Bars, 200 µm. **(C)** Bright-field and fluorescence (AG/BT) images, and FACS analyses for BTAG expression in floating aggregates of the *GATA3*/*SOX17*/*TFAP2C* clone (top) and *GATA2*/*SOX17*/*TFAP2C* clone (bottom) induced with or without Dox 1.0 µg/ml at d4. Representative images for

process (13) (Fig S2A), d6 BT⁺AG⁺ cells from the *GATA3/SOX17/TFAP2C* clone exhibited high similarity/correlation to d2/d4/d6 wild-type BT⁺AG⁺ cells induced by BMP4 (Fig 4B). In contrast, although d2/d4 BT⁺ cells induced from the *GATA3/SOX17* clone appeared to take a similar pathway until d2, they failed to progress further by d4 (Fig S3G).

The numbers of DEGs between BT⁺AG⁺ cells from the *GATA3/SOX17/TFAP2C* clone induced by Dox and from the parental cells induced by BMP4 were the largest at d2 (527), and decreased thereafter (d4: 265; d6: 53) (Fig 4C and Table S2). The genes up-regulated in BMP4-induced cells at day 2 (334 genes) were enriched with those for "negative regulation of transcription from RNA pol II promoter," "embryonic forelimb morphogenesis" (GO terms), and "TGF-β signaling pathway" (KEGG pathway), and included key BMP targets, such as *ID1, ID3, CDX2, TBX3, MSX1, MSX2, HAND1,* and *TFAP2A* (Fig 4D and E), suggesting that these BMP effectors are dispensable for hPGCLC specification. In contrast, the genes up-regulated in *GATA3/SOX17/TFAP2C*-induced cells at day 2 (193 genes) were enriched with those for "negative regulation of BMP signaling pathway," "anterior/posterior axis specification" (GO terms), and "PI3K-Akt signaling pathway" (KEGG pathway), and included *FGF2, FGF12, FGF19, FGFR2, MAP2K1, MAP2K6, IRS2,* and *SOS2* (Fig 4F and G). The genes up-regulated with high fold-changes included epiblast/ectoderm genes such as *SOX2, ZIC3,* and *SALL3,* which were repressed in slower kinetics by the transgene expression (Fig 4G).

We addressed whether *GATA3* and *GATA2* expression were affected by other TFs relevant for hPGCLC specification. As shown in Fig 4H, in any of the knockout clones for *EOMES, SOX17, TFAP2C,* and *BLIMP1* induced for the germ-cell fate by BMP4 (13), *GATA3* and *GATA2* up-regulation was un-affected, indicating that their expression is independent from these TF pathways. We conclude that among the BMP effectors, the GATA TFs are the key that, together with *SOX17* and *TFAP2C,* is sufficient to drive the transcriptional program for hPGCLC specification.

### Critical requirements of the GATA TF paralogs for hPGCLC specification

We next explored whether GATA TFs are essential for hPGCLC induction. Using CRISPR/Cas9 technology (45), we targeted *GATA3* or *GATA2* loci in parental 585B1 BTAG hiPSCs, and isolated four and three clones bearing frameshift mutations in both alleles of *GATA3* or *GATA2,* respectively (*GATA3* or *GATA2* homozygous knockout⁻/⁻ clones) (Fig S4A). The lack of GATA3 or GATA2 expression in these clones was verified by Western blot analyses following the differentiation of these clones into TE-like cells (46) (Fig S4B).

We induced these clones into iMeLCs (Fig S4C), and then into hPGCLCs by BMP4. Unexpectedly, all the *GATA3⁻/⁻* or *GATA2⁻/⁻* clones were induced into BT⁺AG⁺ cells in a manner similar to the parental clone (Figs 5A and S4D and E). We isolated total RNAs from iMeLCs and d2/d4/d6 BT⁺AG⁺ cells induced from all the clones, analyzed the expression of key genes by qRT-PCR, and found that the *GATA3⁻/⁻* or *GATA2⁻/⁻* clones

expressed relevant genes for hPGCLC specification in an apparently normal fashion (Fig 5B). We performed an RNA-seq analysis, which revealed that *GATA3⁻/⁻* and *GATA2⁻/⁻* cells differentiated into BT⁺AG⁺ cells in a manner equivalent to the parental clone (Fig 5C and D), and exhibited small numbers of DEGs compared with the parental counterparts (Fig S4F and G). However, we noted that the BT⁺AG⁺-cell induction efficiencies at d4 of the *GATA3⁻/⁻* clones (~14.5%) were significantly lower than those of the control (~31.4%) or of the *GATA2⁻/⁻* clones (~25.7%) (Fig 5F), raising the possibility that *GATA3* and *GATA2* have a compensatory function, with *GATA3* playing the more dominant role, during hPGCLC induction.

To investigate this possibility, we knocked out the *GATA2* alleles in the *GATA3⁻/⁻* clone, and obtained one line with the *GATA3⁻/⁻; GATA2⁺/⁻* genotype and one line with the *GATA3⁻/⁻; GATA2⁻/⁻* genotype (Fig S4A). Upon the differentiation of these clones into TE-like cells (46), the *GATA3⁻/⁻; GATA2⁺/⁻* cells formed an epithelial-sheet structure with a typical TE-like cobblestone morphology, but the *GATA3⁻/⁻; GATA2⁻/⁻* cells failed to show such differentiation and exhibited a mesenchyme-like appearance (Fig S4H). Consistently, we confirmed that the *GATA3⁻/⁻; GATA2⁻/⁻* cells, but not the *GATA3⁻/⁻; GATA2⁺/⁻* cells, lost the expression of GATA2 proteins (Fig S4B). We then induced these cells into iMeLCs (Fig S4I) and successively to hPGCLCs, which revealed that the *GATA3⁻/⁻; GATA2⁺/⁻* cells still formed the BT⁺AG⁺ cells, but at a further reduced efficiency (~9.5%), whereas the *GATA3⁻/⁻; GATA2⁻/⁻* cells barely differentiated into such a state (~1.6%) (Fig 5E and F).

To exclude the possibility that the differentiation failure of the *GATA3⁻/⁻; GATA2⁻/⁻* clone was due to a clonal effect, we performed a rescue experiment. We introduced the Dox-inducible *GATA3* expression system into the *GATA3⁻/⁻; GATA2⁻/⁻* clone and isolated a line that showed an appropriate *GATA3* expression in hiPSCs. We induced this line into iMeLCs (Fig S4I), and stimulated the iMeLC aggregates with BMP4 and Dox. Although we found that a continuous stimulation of the iMeLC aggregates with BMP4 and Dox led to major cell death for an unknown reason, the timed stimulation of Dox (~32 h) resulted in the induction of BT⁺AG⁺ cells in a Dox-dose dependent manner (Fig 5G–I), and the induced BT⁺AG⁺ cells expressed key genes for hPGCLCs in an appropriate fashion (Fig 5J). Thus, we conclude that the GATA TF paralogs, *GATA3* and *GATA2,* show a dose-dependent requirement for hPGCLC specification. Considering that *GATA3* was expressed at a higher level than *GATA2* during hPGCLC induction (Fig 3A) and upon cyPGC specification (Fig 2I) and that *GATA3* knockouts, but not *GATA2* knockouts, exhibited a significant decrease in hPGCLC induction efficiency (Fig 5F), we propose that *GATA3* plays a major role in hPGCLC induction.

### GATA3 expression in post-implantation primate embryos

To explore the spatial relationship of GATA3, SOX17, and TFAP2C expression in a developmental context, we examined their expression during PGC specification in the early post-implantation embryos of cynomolgus monkeys. By immunofluorescence analysis, we

---

10 (*GATA3/SOX17/TFAP2C*) and six (*GATA2/SOX17/TFAP2C*) experiments are shown. Bars, 200 μm. **(D)** Expression of *rBGpA* (transgenes) and the indicated endogenous genes in BMP-induced parental clone-derived and Dox-induced *GATA3/SOX17/TFAP2C* clone- and *GATA2/SOX17/TFAP2C* clone-derived d4 BT⁺AG⁺ cells. Two replicates from independent experiments were analyzed. For each gene, the ΔCt was calculated from the average Ct value of two housekeeping genes, *RPLP0* and *PPIA* (set as 0). **(E, F)** Bright-field and fluorescence (AG/BT) images, and FACS analyses for BTAG expression at d4 in floating aggregates of the *GATA3/SOX17/TFAP2C* (E) and *GATA2/SOX17/TFAP2C* (F) clones induced by BMP4 or Dox with 0, 100, 500 nM of LDN193189. Representative images of at least two independent experiments are shown. Bars, 200 μm.

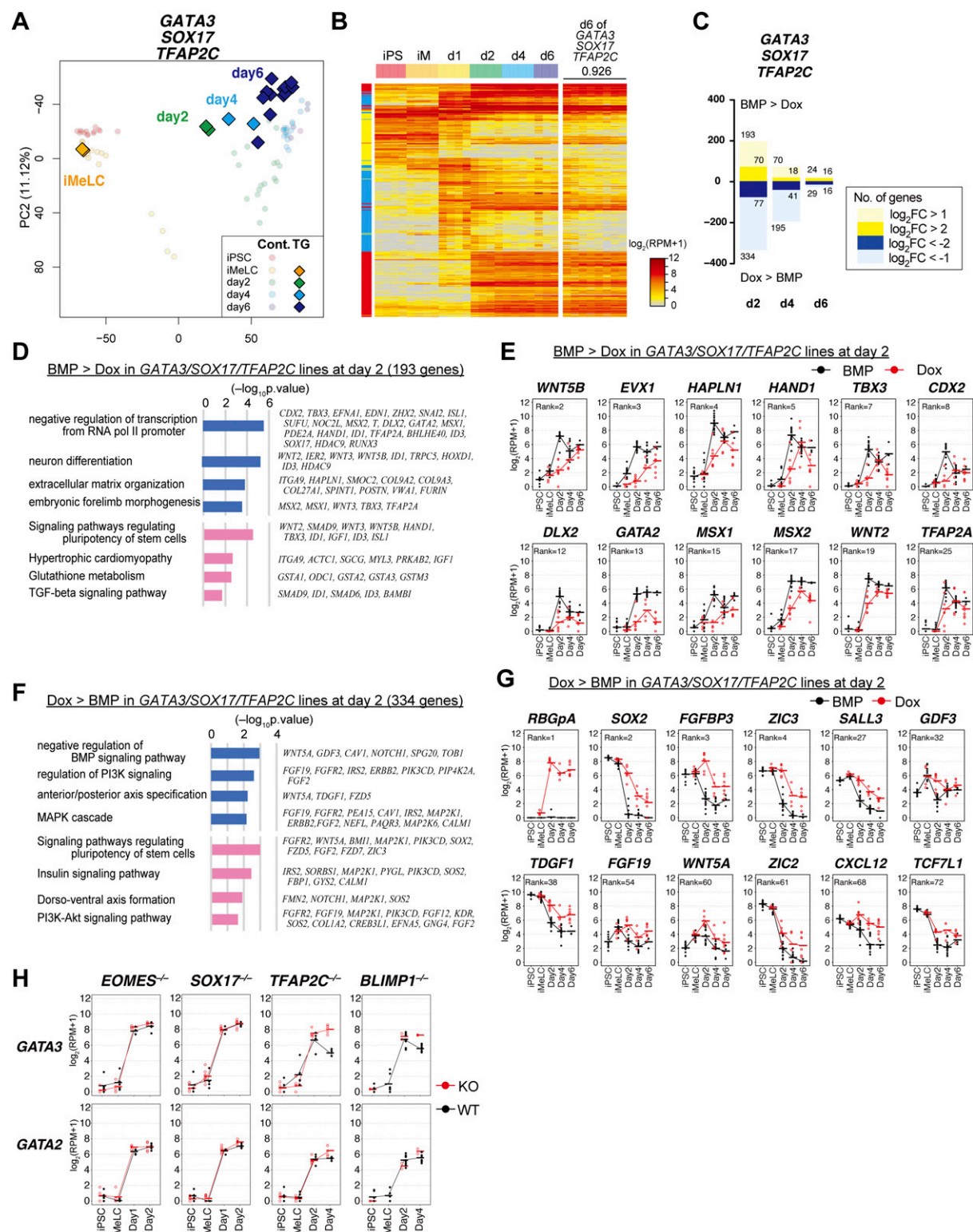

**Figure 4.   The transcription factor-induced BT⁺AG⁺ cells directly acquire human primordial germ-cell-like cell (hPGCLC) transcriptome.**
**(A)** The principal component analysis plots of the cells (iMeLCs, Dox-induced d2/d4/d6 BT⁺AG⁺ cells) derived from the *GATA3/SOX17/TFAP2C* clone (squares), overlaid with the indicated cells derived from the parental clone (circles with pale color). See Table S1 for the samples analyzed. The color coding is as indicated. **(B)** Heat map representation (color coding as indicated) of the expression of the 481 genes characterizing hPGCLC specification (13) (Fig S2) in the parental hiPSCs, iMeLCs, d1 whole aggregates, and d2/d4/d6 BT⁺AG⁺ cells and in Dox-induced, *GATA3/SOX17/TFAP2C*–derived d6 BT⁺AG⁺ cells. The correlation coefficient (0.926) between bone morphogenetic protein (BMP)– and Dox-induced d6 BT⁺AG⁺ cells is shown. The color coding in the left column is as follows: red, genes for PGCLC specification; cyan, genes

detected SOX17[+] and TFAP2C[+] cyPGCs in the dorsal amnion at E12, in the posterior amnion at E13, and near the primitive streak region between the epiblast and hypoblast at E15 (Fig 6A–D). During this period, SOX17 was also expressed in the hypoblast, and TFAP2C was also expressed diffusely in the amnion and cytotrophoblast/syncytiotrophoblast at E12 but became negative in the amnion thereafter (Fig 6A–C). We found that GATA3 was expressed strongly in the cytotrophoblast/syncytiotrophoblast and also in the amnion and the hypoblast, and importantly, in the SOX17[+]/TFAP2C[+] cyPGCs (Fig 6A–D). Along with the embryonic development, we noted a gradual decrease in the ratio of GATA3[+] cyPGCs (Fig 6D). Notably, the epiblast did not express any of these TFs.

In addition, we re-analyzed *GATA3/GATA2* expression in the single-cell transcriptome of cy post-implantation embryos (E13–E20) (30). In addition to early cyPGCs (Fig 6A–D), *GATA3* was expressed in the TE, extra-embryonic mesenchyme, visceral endoderm/yolk-sac endoderm, and gastrulating cells, and weakly/sporadically in the epiblast (Fig 6E). *GATA2* was expressed strongly in the TE, but was weak/sporadic in the other cell types, including early cyPGCs (Figs 2J and 6E). These findings delineate the spatial relationship of the expression of GATA3, SOX17, and TFAP2C during primate development, demonstrating that cyPGCs, but not other relevant cell types, co-express these TFs.

### TF-induced hPGCLCs are competent for epigenetic reprogramming and differentiation into oogonia/gonocytes

To evaluate the functional property of the BT[+]AG[+] cells induced by *GATA3*, *SOX17*, and *TFAP2C* as human germ cells, we explored whether they are competent to undergo epigenetic reprogramming and to differentiate into oogonia/gonocytes, which are female/male gonadal germ cells before overt sexual differentiation and exhibit similar gene-expression and epigenetic profiles (47, 48, 49). Both female and male hiPSCs differentiate into oogonia/gonocytes (10). Accordingly, for this purpose, we isolated the Dox-induced *GATA3/SOX17/TFAP2C* clone-derived d6 BT[+]AG[+] cells, and aggregated them with mouse embryonic ovarian somatic cells at E12.5 to form xenogeneic reconstituted ovaries (xrOvaries) in vitro (10, 11) (Fig 7A). As a control, we generated xrOvaries using the BMP4-induced parental clone-derived d6 hPGCLCs. We isolated xrOvaries at d77 of aggregation culture (ag77) and evaluated them with im-munofluorescence analyses. The analyses revealed that multiple clusters of *GATA3/SOX17/TFAP2C* clone-derived AG[+]/human mito-chondrial-antigen[+] cells persisted in xrOvaries, and many of them expressed DDX4, a key marker of oogonia/gonocytes (Fig 7B).

To further characterize the *GATA3/SOX17/TFAP2C* clone-derived BT[+]AG[+] cells at ag77, we isolated them by FACS, determined their transcriptome by an RNA-sequence, and analyzed their properties. PCA revealed that both the *GATA3/SOX17/TFAP2C* clone-derived cells and the parental d6 hPGCLC-derived cells (generated in the present study) differentiated as human germ cells in an appro-priate manner, acquiring the transcriptome property of oogonia/gonocytes (Fig 7C). Accordingly, the *GATA3/SOX17/TFAP2C* clone-derived cells expressed key markers of oogonia/gonocytes, in-cluding *DPPA3*, *DAZL*, *DDX4*, and *MAEL*, at high levels (Fig 7D), and with regard to the expression of 451 genes that characterize the oogonia/gonocyte differentiation process from hiPSCs (10), the *GATA3/SOX17/TFAP2C* clone-derived cells exhibited profiles equivalent to those of the oogonia/gonocytes (Fig S5). The finding that both the *GATA3/SOX17/TFAP2C* clone-derived cells and the control cells generated in this study were plotted at positions closest to the previous ag63 cells rather than ag77 cells in the PCA (Fig 7C) would be attributable to experimental variation.

Epigenetic reprogramming, including genome-wide DNA deme-thylation, is a hallmark and exclusive event during germ-cell de-velopment (4, 6, 7). To examine whether the *GATA3/SOX17/TFAP2C* clone-derived BT[+]AG[+] cells at ag77 undergo epigenetic reprog-ramming, we determined their genome-wide DNA methylation profile by whole-genome bisulfite sequence (WGBS) analysis. Im-portantly, we found that the genome-wide DNA methylation properties of the *GATA3/SOX17/TFAP2C* clone-derived cells at ag77 were similar to those of oogonia/gonocytes reported previously (10,47,50) with regard to both their distribution profiles (Fig 7E) and total levels (reduced to as low as ~17.5%) (Fig 7F). Accordingly, the *GATA3/SOX17/TFAP2C* clone-derived cells erased their DNA meth-ylation throughout their genomic regions, including promoters, exons, introns, intergenic regions, and non-promoter CpG islands (CGIs) (Fig 7G), as well as parental imprint control regions (ICRs) (Fig 7H). Collectively, these findings demonstrate that the BT[+]AG[+] cells induced by *GATA3*, *SOX17*, and *TFAP2C* are equivalent to hPGCLCs in their capacity to undergo epigenetic reprogramming and to dif-ferentiate into oogonia/gonocytes, and thus are considered to bear key properties of bona fide hPGCs.

## Discussion

We have identified core TFs, GATA3/2, SOX17, and TFAP2C, which suffice for the reconstitution of the human germ-cell fate, providing a step forward for delineating the mechanism of human germ-cell specification and a foundation for the TF-based human gameto-genesis (Fig 7I). Our finding that *SOX17*, *TFAP2C*, and *BLIMP1*

---

for endoderm/mesoderm specification; yellow, genes for pluripotency. See Fig S2 for details. **(C)** The numbers of differentially expressed genes between BMP- and Dox-induced cells of the *GATA3/SOX17/TFAP2C* clone in d2/d4/d6 BT[+]AG[+] cells ($P < 0.01$ by Tukey–Kramer test, $\log_2$[RPM + 1] > 4 in cells with higher expression, $\log_2$[fold change: FC] > 1 [up, pale yellow; down, pale blue] or 2 [up, yellow; down, blue]). **(D, F)** Gene ontology terms (blue) and KEGG pathways (pink) enriched in genes up-regulated in BMP-induced (D) and in Dox-induced (F) *GATA3/SOX17/TFAP2C* clone-derived d2 BT[+]AG[+] cells. **(E, G)** Expression dynamics of the genes up-regulated in BMP-induced (E, the parental clone) or Dox-induced (G, the *GATA3/SOX17/TFAP2C* clone) d2 BT[+]AG[+] cells during the respective induction processes (BMP: black; Dox: red). The bars indicate the mean value of each time point, and the rank of the gene ordered by the fold change is shown. See Table S1 for the samples analyzed. **(H)** Expression dynamics of *GATA3* and *GATA2* during BMP-induced hPGCLC induction from the parental (gray) and the indicated knockout hiPSCs (red) (d1: iMeLC aggregates; d2: *EOMES*[−/−]: whole aggregates; *SOX17*[−/−]: whole aggregates; *TFAP2C*[−/−]: BT[+] cells; *BLIMP1*[−/−]: AG[+] cells) (13). The bars indicate the mean value of each time point of each genotype. See Table S1 for the samples analyzed.

Figure 5. Dose-dependent function of GATA transcription factors in human primordial germ-cell-like cell specification.

**(A)** Bright-field and fluorescence (AG/BT) images, and FACS analyses for BTAG expression in floating aggregates of the *GATA3*^−/−^, *GATA2*^−/−^, and parental clones induced by bone morphogenetic protein 4 (BMP4) at d2/d4. Representative images of at least two independent experiments are shown (indicated in Fig 5F). Bars, 200 μm.
**(B)** Expression dynamics of the indicated genes during human primordial germ-cell–like cell induction (iMeLCs, d2/d4 BT⁺AG⁺ cells) by BMP4 from the parental (black), *GATA3*^−/−^ (red), and *GATA2*^−/−^ (blue) clones. For each gene, the ΔCt was calculated from the average Ct value of two housekeeping genes, *RPLP0* and *PPIA* (set as 0). The bars indicate the mean value of each time point of each genotype. Replicate numbers: *GATA2*^−/−^: 3; *GATA3*^−/−^: 4 for iMeLCs and 8 for d2/d4 BT⁺AG⁺ cells; parental clone: 2.

expression in iMeLCs was not sufficient to induce hPGCLCs was unexpected, given that in mice, the expression of three TFs, *Blimp1*, *Prdm14*, and *Tfac2c*, and with a lesser efficiency, two TFs (*Blimp1* and *Tfap2c*; *Prdm14* and *Tfap2c*) or even a single TF (*Prdm14*), was sufficient to confer the germ-cell fate on their precursors (25). Notably, none of these TFs are known as direct effectors of the BMP signaling in mice; *TFAP2C* and *BLIMP1* are shared by humans; and *PRDM14* is expressed in both hiPSCs and iMeLCs (9, 16). Thus, the mechanism of human germ-cell specification is distinct from that in mice not only with regard to the TFs themselves and their hierarchies of actions (13), but also in terms of how the key TFs regulate each other to drive the downstream PGC pathways.

We showed that *SOX17* induces *BLIMP1* in both hiPSCs and iMeLC aggregates (Fig 1B and E), and the *TFAP2C/BLIMP1*, *SOX17/TFAP2C*, *SOX17/TFAP2C/BLIMP1* clones induce *NANOS3*, a marker for hPGCLCs (Fig 1B). Combined with the finding that the *BLIMP1^{-/-}* clones failed to up-regulate *NANOS3* (9, 13), these data indicate that *NANOS3* is most likely a downstream target of *BLIMP1*. Thus, *SOX17* induces *BLIMP1*, which in turn induces *NANOS3*; however, this is evidently insufficient for hPGCLC specification (Fig 1E–H). The addition of *TFAP2C* expression to the *SOX17/BLIMP1/NANOS3* pathway also failed to activate the hPGCLC specification program (Fig 1E), indicating that although both *SOX17* and *TFAP2C* are essential and interdependent for hPGC(LC) specification (13), the two pathways do not activate each other to elicit the hPGC(LC) specification program.

We identified two GATA TFs—*GATA3* and *GATA2*, with *GATA3* playing the more dominant role—as key BMP effectors that, together with *SOX17* and *TFAP2C*, drive the hPGC(LC) specification program (Figs 3C–F and 7I). Importantly, *GATA3* and *SOX17* expression in iMeLCs was not sufficient to induce the hPGCLC program (Fig S3E), suggesting that the expression of three TFs, *GATA3/2*, *SOX17*, and *TFAP2C*, is a minimal requirement in replacing the BMP signaling and conferring the germ-cell fate on iMeLCs. This would in turn suggest that in humans, the BMP signaling plays a key role in activating not only *GATA3/2*, but also, directly or indirectly, *SOX17* and *TFAP2C*, because BMP-induced *GATA3/2* per se was not sufficient to activate *SOX17* and *TFAP2C*. The mechanisms by which the three TFs control each other as well as the downstream pathways for hPGCLC specification require further investigation. In this regard, it is interesting to note that *GATA3* is required only transiently for hPGCLC induction (Fig 5G–J), and that GATA TFs are known to act as "pioneering factors" that open heterochromatin regions and make them accessible to other TFs (51, 52, 53). Accordingly, it has been reported that hPGCLCs and human fetal germ cells bear open chromatin regions enriched in the binding motifs for OCT4, SOX/TFAP/KLF families, and GATA families as well (15), and that the

expression of *GATA3* as well as *TFAP2A* precedes that of *SOX17* and *TFAP2C* at a single-cell level (54), supporting the idea that GATA TFs may operate as "pioneering factors" for other TFs such as SOX17 and TFAP2C to drive germ-cell specification and thereafter to maintain germ-cell identity (Fig 7I).

A number of studies have shown that in response to BMP4, hPSCs differentiate into TE-like cells with the expression of key TFs, including *GATA3*, *GATA2*, *TFAP2A*, and *TFAP2C* (39, 46, 55, 56), and thus the differentiation of hPSCs into TE-like cells involves signals and TFs that are also involved in hPGCLC specification (this study) (54). These TFs are also associated with the TE development in mice (40, 57, 58), and as in the case of hPGCLC specification, *GATA3* and *GATA2* play a compensatory function for TE-like cell specification in humans (39) and TE development in mice (40). A characteristic gene involved in hPGC(LC) differentiation is *SOX17*, the activation of which requires stimulation of hPSCs by WNT signaling that activates *EOMES* expression (i.e., iMeLC induction), before stimulation by BMP signaling (13). In contrast, TE-like cell differentiation requires direct stimulation of hPSCs by BMP signaling (39, 55, 56) and such differentiation has been shown to be significantly promoted by inhibiting endogenous WNT activity elicited by BMP (59). Thus, evidently, a prior activation of the WNT pathway in hPSCs is a key to the differential cell-fate specification between the germ-cell fate and TE-like cell fate upon BMP stimulation.

Notably, *GATA3*, *GATA2*, *TFAP2A*, and *TFAP2C* are also expressed in the amnion in cynomolgus monkeys (Fig 6A–C) (18) and most likely in humans (60, 61), indicating that relatively close lineage relationships exist among TEs, amnion and PGCs in primates. Indeed, during primate development, TEs are specified from the inner cell mass (ICM) cells of the pre-implantation blastocysts (around embryonic day [E] 4 in humans and E5 in cy monkeys) 30, 62, 63, 64, and subsequently, the amnion is differentiated from ICM/epiblasts around the peri-implantation stage (around E7 in humans and E11 in cy monkeys) 18, 30, 62, 63, 64 and the PGCs are most likely originated in the nascent amnion (unknown for humans and E11 in cy monkeys) (18, 54). Thus, TEs, amnion and PGCs are the lineages that arise successively from the ICM/epiblast during the relatively short period of early development. On the other hand, the transcriptome of h/cyPSCs is highly similar to those of post-implantation early (E12/13) or late (E16/17) epiblast cells, and is substantially different from those of ICM cells or pre-implantation epiblast (30), making it difficult to naturally reconcile the observation that hPSCs bear a capacity to differentiate into TE-like cells; further investigations will be needed to account for this apparent paradox. The mechanism that segregates the germ-cell fate from the amnion fate, which also responds to WNT signaling (18), also remains an open question, and an

---

**(C, D)** The principal component analysis plots of the cells (iMeLCs, d2/d4/d6 BT^+AG^+ cells) derived from the *GATA3^{-/-}* (C) and *GATA2^{-/-}* (D) clones (squares), overlaid with the indicated cells derived from the parental clone (circles with pale colors). See Table S1 for the samples analyzed. The color coding is as indicated. **(E)** Bright-field and fluorescence (AG/BT) images, and FACS analyses for BTAG expression in floating aggregates of the *GATA3^{-/-}*; *GATA2^{+/-}* and *GATA3^{-/-}*; *GATA2^{-/-}* clones induced by BMP4 at d4. Representative images of at least two independent experiments are shown (indicated in Fig 5F). Bars, 200 μm. **(F)** The percentages of BT^+AG^+ cell induction from the indicated genotypes at d4. The replicate numbers and the *P*-values (*t* test) are as indicated. The inductions were performed side by side. Typically, the efficiency for BT^+AG^+ cell induction from parental hiPSCs varies to this extent (20%~60%) (9, 13, 26). **(G)** A scheme for *GATA3* expression in the *GATA3^{-/-}*; *GATA2^{-/-}*; *GATA3* clone. **(H)** Bright-field and fluorescence (AG/BT) images, and FACS analyses for BTAG expression in floating aggregates of the *GATA3^{-/-}*; *GATA2^{-/-}*; *GATA3* clone upon induction with BMP4 and 0, 0.5, and 1.0 μg/ml of Dox at d4. Representative images of at least two independent experiments are shown (indicated in Fig 5F). Bars, 200 μm. **(I)** The percentages of BT^+AG^+ cell induction at d4 (two replicates) from the *GATA3^{-/-}*; *GATA2^{-/-}*; *GATA3* clone induced with BMP4 and 0, 0.5, and 1.0 μg/ml of Dox treatment. **(J)** Expression of the indicated genes in d4 BT^+AG^+ cells induced from the parental (with BMP4, red) or *GATA3^{-/-}*; *GATA2^{-/-}*; *GATA3* (with BMP4 and Dox, green) clones (two replicates). For each gene, the ΔCt was calculated from the average Ct value of two housekeeping genes, *RPLP0* and *PPIA* (set as 0).

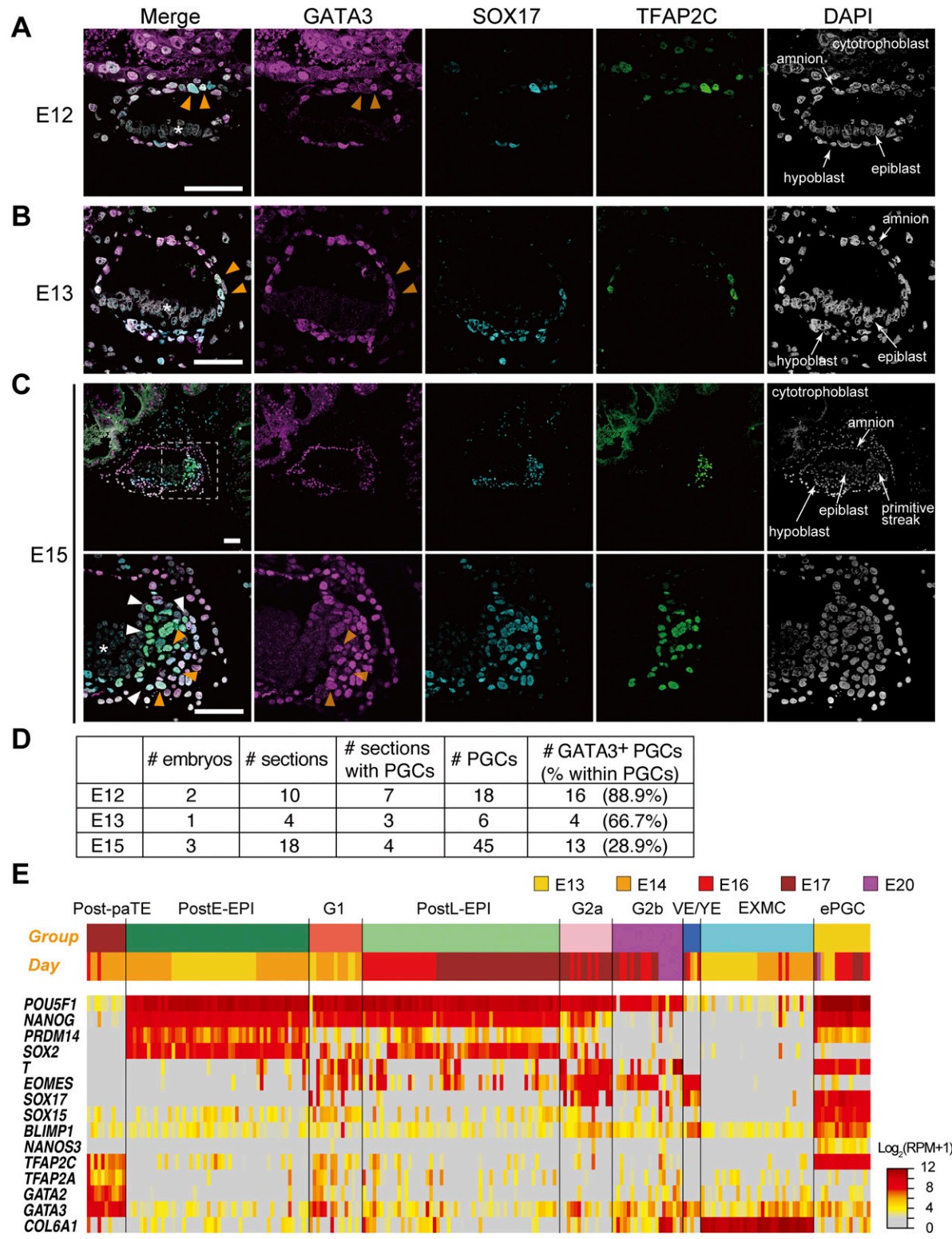

**Figure 6. GATA3 expression in post-implantation cynomolgus monkey embryos.**
**(A, B, C)** Immunofluorescence analysis of GATA3 (magenta), SOX17 (cyan), and TFAP2C (green) expression (merged with DAPI) in cynomolgus monkey embryos at E12 (A), E13 (B) and E15 (C). For E15 (C), the boxed area (top) is magnified (bottom). Orange arrowheads indicate triple-positive PGCs, whereas white arrowheads indicate SOX17/TFAP2C+ but GATA3- cells. Note that there was no signal in the epiblast (asterisk). **(D)** Representative images of the samples shown in (D) are presented. Bars, 50 µm. **(D)** The numbers of embryos (E12, E13, and E15), total sections, and sections with PGCs (SOX17+TFAP2C+ cells), PGCs, and GATA3+ PGCs analyzed/detected in this study. **(E)** Heat map representation of the expression of key genes in the indicated cell types of the post-implantation cy embryos (30). The colored bars on top indicate cell types (top)

understanding of this mechanism may lead to a more efficient induction of the germ-cell fate from hPSCs.

Crucially, we showed that the TF-induced BT⁺AG⁺ cells, when cultured in xrOvaries, underwent epigenetic reprogramming and differentiated into oogonia/gonocytes (Fig 7), demonstrating that the TF-induced BT⁺AG⁺ cells bear one of the key functions of bona fide hPGCs. Unlike mPGCLC specification, which is directly coupled with epigenetic reprogramming (65, 66, 67, 68), hPGCLC specification itself does not appear to be sufficient to elicit the epigenetic reprogramming: further signaling/environmental cues, including those provided by xrOvaries, are necessary to activate such key processes (10, 50). Upon mPGCLC specification, Blimp1, Prdm14, and Tfap2c repress the expression of genes such as *Dnmt3a/b* and *Uhrf1*, and create a cellular state with little, if any, de novo and maintenance DNA methyltransferase (DNMT) activities (50, 65, 66, 67, 68), and this leads to a replication-coupled passive genome-wide DNA demethylation upon mPGCLC proliferation (67, 68). In contrast, the mechanism of epigenetic reprogramming, including genome-wide DNA demethylation, in humans is unclear, and may involve a divergence from that in mice. The identification of the TFs sufficient to create the hPGCLC state (this study), coupled with the development of a method for hPGCLC expansion (50), will be instrumental in clarifying the mechanism of epigenetic reprogramming in human germ cells.

The mechanisms of germ-cell specification in metazoans are classified largely into two modes, "epigenesis" and "preformation" (69, 70). The former, as in mammals, involves a strategy to induce the germ-cell fate into pluripotent precursors by signaling molecules and is evolutionarily ancestral, whereas the latter, as in flies and frogs, involves "preformed" germ plasm in oocytes for germ-cell specification and has been acquired in diverse metazoan lineages as a result of convergent evolution (69, 70). Notably, in the "epigenesis" mode, BMP has been identified as an evolutionarily conserved key signal in species as diverse as gryllus (71), axolotl (72), and mammals, including mice (73, 74), rabbits (75), pigs (76), monkeys (18, 19), and humans (8, 9). On the other hand, there has been a lack of knowledge as to the mechanism of action, including via direct effectors, of the BMP signaling for PGC specification in these species. In future investigations, it would be useful to investigate whether GATA TFs—which are widely evolutionarily conserved—play a similar role in diverse species, including mice.

# Materials and Methods

## Animal care and use

All animal experiments were performed under the ethical guidelines of Kyoto University and Shiga University of Medical Science. Pregnant ICR female mice were purchased from Japan SLC. Experimental procedures using cynomolgus monkeys were approved by the Animal Care and Use Committee of the Shiga University of Medical Science.

## Human iPSC culture

All the experiments on the induction of hPGCLCs from hiPSCs and genome editing were approved by the Institutional Review Board of Kyoto University and were performed according to the guidelines of the Ministry of Education, Culture, Sports, Science, and Technology (MEXT) of Japan.

The 585B1 BTAG hiPSCs (46, XY) (9) were maintained in StemFit AK03N medium (Ajinomoto) on cell culture plates coated with iMatrix-511 (Nippi) (77). The medium was changed every other day. For the passage or the induction of differentiation, the cells were treated with a 1 to 1 mixture of TrypLE Select (Life Technologies) and 0.5 mM EDTA/PBS to dissociate into single cells, and 10 µM of a ROCK inhibitor (Y-27632; Wako Pure Chemical Industries) was added for 24 h after plating.

## Generation of TF-expression lines

The vectors for the Doxycycline-induced expression were constructed based on the Gateway System (Thermo Fisher Scientific) as described previously (13). The full-length cDNA sequences of *SOX17*, *TFAP2C*, *BLIMP1*, *MSX2*, *GATA3*, and *GATA2* were PCR amplified from d2 hPGCLCs derived from the 585B1 BTAG hiPSC line. Nucleotide sequences for the epitope tags with linkers, 3×FLAG-G4S, V5-G4S, and 2×TY1-G4S, were added to the 5-prime ends of *SOX17*, *TFAP2C*, and *BLIMP1/MSX2/GATA3/GATA2*, respectively. Primers used for the construction are shown in Table S3. The PCR products were cloned between the BamHI and XhoI sites of the pENTR1a vector and were subsequently recombined into the destination vector with LR clonase. In the destination vector, the transgenes were cloned under the TetO promoter repeat region and followed by the rabbit β-globin poly A (rβGpA) sequence. In the region downstream of *rβGpA*, the puromycin-resistant gene driven by the *EF1α* promoter was cloned for *SOX17* and *TFAP2C*, neomycin for rtTA, and hygromycin for the rest of the genes.

Transfection was performed with the electroporator NEPA21 type II (Nepagene). Half a million hiPSCs were transfected with 500 ng of the *piggybac* transposase expression vector and 1 µg of each transgene expression vector, except for TFAP2C, which was added at 1.5 µg, then resuspended in 100 µl of OptiMEM (Thermo Fisher Scientific). Selection antibiotics (200 µg/ml geneticin [G418], 10 µg/ml puromycin, and 800 µg/ml hygromycin [all from Thermo Fisher Scientific]) were added 2 d after the transfection and maintained until the surviving colonies were picked up at 12–14 d. The induction of the transgenes with 1.0 µg/ml doxycycline (Takara-Clontech) for the selected hiPSC clones was assessed at 24 h of culture.

## Generation of knockout lines

pX335-U6-Chimeric BB-CBh-hSpCas9n (D10A) was a gift of Feng Zhang (plasmid #42335; Addgene) (45), and the *eGFP* sequence was replaced with the mCherry sequence bearing a silent mutation

and embryonic days (bottom), respectively. The color coding is as indicated. Post-paTE, PostE-EPI, post-implantation early epiblast; G1, gastrulating cells, group 1; postL-EPI, post-implantation late epiblast; G2a/G2b, gastrulating cells, group 2a/2b; VE/YE, visceral endoderm/yolk sac endoderm; EXMC, extraembryonic mesenchyme; ePGC, early PGC.

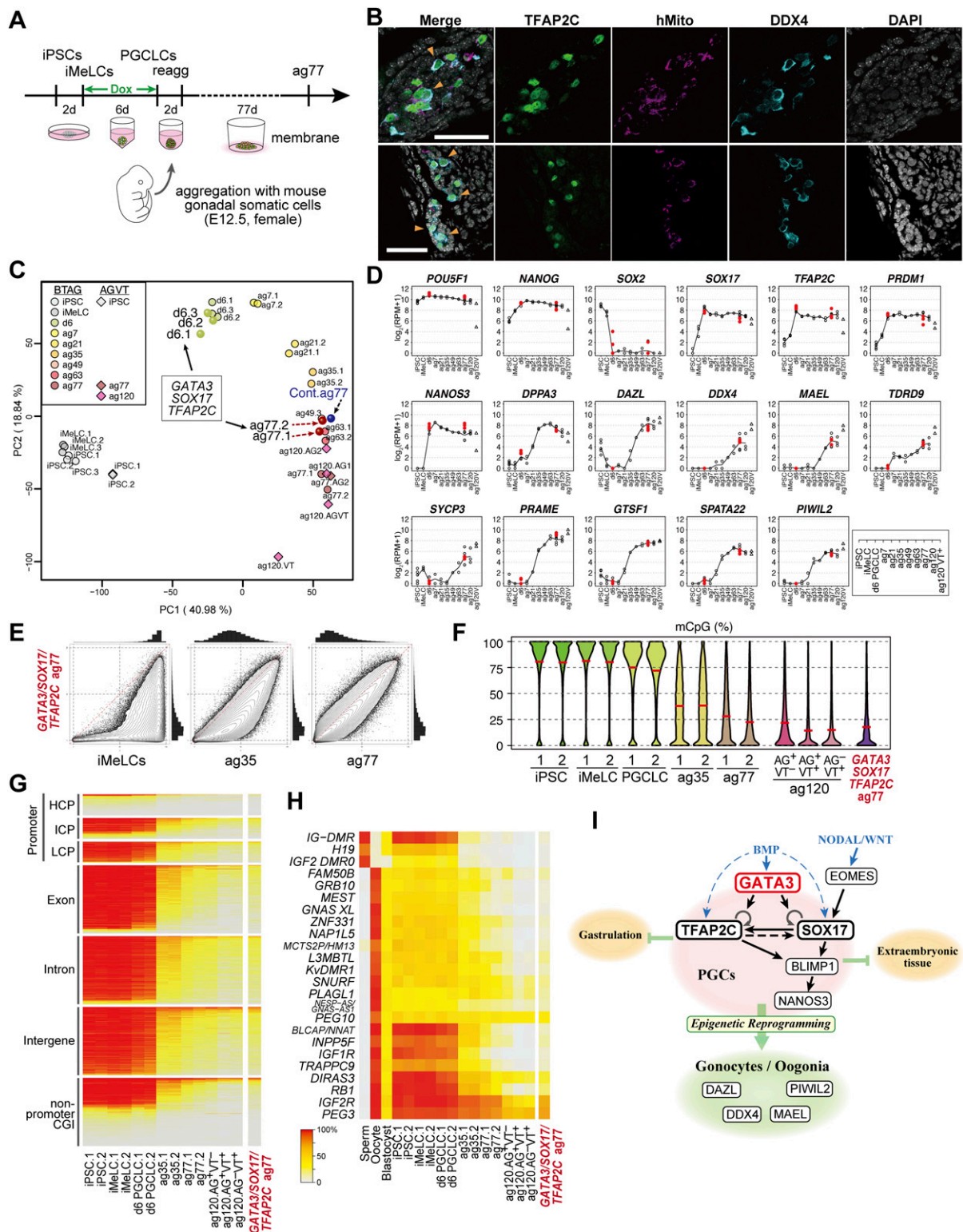

**Figure 7. Transcription factor-induced BT⁺AG⁺ cells differentiate into oogonia/gonocytes in xrOvaries.**
**(A)** A scheme for xrOvary culture (10, 11) with Dox-induced *GATA3/SOX17/TFAP2C* clone-derived d6 BT⁺AG⁺ cells. **(B)** Immunofluorescence analysis of TFAP2C (green),
human mitochondria antigen (magenta), and DDX4 (cyan) expression (merged with DAPI) on aggregation day (ag) 77 xrOvaries. In two independent experiments, 28 and 23
TFAP2C/DDX4-expressing cells/7 sections, respectively, were detected. Bars, 50 μm. **(C)** The principal component analysis plots of the transcriptome of the *GATA3/SOX17/
TFAP2C* clone-derived d6 BT⁺AG⁺ cells, d6ag77 BT⁺AG⁺ cells, and the parent clone-derived bone morphogenetic protein-induced d6ag77 BT⁺AG⁺ cells (see Table S1)
with the relevant cell types during in vitro oogonia/gonocyte differentiation reported in reference 10, in which 585B1 BTAG hiPSCs (XY) and 1390G3 AGVT (AG; *DDX4* [also

(G432A) for convenience in clone selection. For one recombination site, a pair of gRNA sequences nicking each strand of genomic DNA were designed using a CRISPR design website (crispr.mit.edu/: now renovated). Two oligo DNAs bearing the gRNA sequence and compatible ends for the BbsI-digested overhang sequence were phosphorylated, annealed and ligated into the BbsI site of the pX335 vector. 2 $\mu$g each of the CRISPR vectors in 100 $\mu$l of OptiMEM were transfected into 5 × 10$^5$ 585B1 BTAG hiPSCs using NEPA21 type II (Nepagene). The cells were cultured in AK03N with 10 $\mu$M Y27632 for 24 h, then maintained in AK03N alone for the next 24 h, and the cells with high mCherry expression (≈top 0.5%) were plated onto 96-well plates precoated with iMatrix at a single cell per well with the Automatic Cell Deposition Unit of the FACS Aria III (BD Biosciences). The cells were then cultured in AK03N with 10 $\mu$M Y27632 for 72 h and subsequently cultured in AK03N alone. 10–14 d after plating, pro-liferating colonies were collected: half of the cells were frozen in Stem CellBanker (Zenoaq) and the remaining half were pelleted and lysed for genotyping.

### Genotype

To extract the genomic DNA, the cells were lysed in 40 $\mu$l of KOD Plus Neo buffer (TOYOBO) supplemented with 0.5% NP40 and 0.8 mg/ml Proteinase K (TakaraBio) at 55°C for 3 h, followed by a proteinase inactivation step at 95°C for 10 min. PCR amplification at the CRISPR recombination site was performed from 1 $\mu$l of the cell lysate with KOD Plus Neo (TOYOBO) and the primers listed in Table S3. To sequence each allele separately, the amplicon was A-tailed with Taq polymerase (Greiner), cloned into pGEM-T easy (Promega), and transformed into the DH5$\alpha$ *Escherichia coli* strain, followed by plating onto LB plates with a blue-white selection. White colonies were picked for direct PCR with KOD FX Neo (TOYOBO) using the primers M13-RV and M13-M4. The amplified fragments were se-quenced by Eurofins Genomics with the M13-FW primer and searched for insertions and/or deletions.

### Western blot analysis

Cell pellets were lysed in Bolt Sample Buffer Reducing Agent and the protein concentrations were measured by protein quantifica-tion assay (Macharey-Nagel). Bolt 10% Bis-Tris Plus Gels were used for SDS–PAGE with 1 $\mu$g of samples per lane and subsequently transferred to Polyvinylidene difluoride (PVDF) membrane (0.22 $\mu$m pore) with an iBlot2 Dry Blotting System. All the reagents and devices were from Thermo Fisher Scientific if not specified and all the experiments were performed following the manufacturer's instructions. The transferred membrane was washed in PBST (PBS with 0.1% Tween20), blocked with the blocking solution (5% skim milk in PBST) for 20 min at room temperature with continuous rocking and incubated overnight at 4°C with primary antibodies diluted at 1:1,000 in blocking solution. After washing three times × 5 min with PBST, the membrane was incubated with the HRP-conjugated secondary antibodies diluted at 1:1,000 in blocking solution for 1–3 h at room temperature. After washing three times × 5 min with PBST, the chemiluminescent reaction was induced with Amersham ECL Western Blotting Detection Reagent (GE Healthcare Life Sciences) and the signal was detected with Fusion Solo S (Vilber). All the antibodies used are listed in the Reagents and Tools table.

### hPGCLC induction

The induction of hPGCLCs via iMeLCs was performed as described previously (9, 13, 26). For the induction of iMeLCs, hiPSCs were plated at a density of 5 × 10$^4$ cells/cm$^2$ onto a fibronectin (FC010; Millipore)-coated plate. Either 24-well, 12-well or 6-well plates were used according to the number of cells required. The cells were cultured in GK15 medium (GMEM with 15% KSR, 0.1 mM NEAA, 2 mM L-glu-tamine, 1 mM sodium pyruvate, penicillin-streptomycin, and 0.1 mM 2-mercaptoethanol) supplemented with 50 ng/ml activin A (R&D Systems), 3 $\mu$M CHIR99021 (TOCRIS), and 10 $\mu$M of Y-27632 (Wako Pure Chemical Industries) for 44–48 h. Then the cells were disso-ciated into single cells with TrypLE Select and aggregated in a low-cell-binding V-bottom 96-well plate (Greiner) at 5,000 cells per well in 100 $\mu$l of GK15 medium supplemented with 200 ng/ml BMP4 (R&D Systems), 100 ng/ml SCF (R&D Systems), 50 ng/ml EGF (R&D Sys-tems), 1,000 U/ml LIF (Millipore), and 10 $\mu$M of Y-27632 to be induced into hPGCLCs. For the transgene-mediated induction, BMP was replaced with doxycycline (Dox) at 1.0 $\mu$g/ml except in the case of *GATA2* overexpression, for which 0.5 $\mu$g/ml Dox was used. The medium was not changed until the analysis up to the sixth day of induction except in the case of the *GATA3* rescue experiment. For the *GATA3* rescue experiment, 100 $\mu$l of medium containing both Dox and BMP4 was applied first, and then, after 32 h, 90 $\mu$l of the medium was aspirated and replaced with the same amount of medium containing BMP4, SCF, EGF and LIF, and the culture was continued for the remaining days. The images of the aggregates were taken under an M205C stereo microscope (Leica Micro-systems) equipped with a DP72 CCD camera and DP2-BSW software (Olympus).

### FACS

For analysis of the cellular contents of the aggregates, the ag-gregates were collected on the designated days of induction, washed once in PBS, and dissociated with 0.25% Trypsin–EDTA for

known as *VASA*]-*tdTomato* [VT]) hiPSCs (XX) were used as starting materials. Numbers following "ag" indicate the culture days in xrOvaries. For the AGVT cells, ag77 and 120 AG$^+$VT$^-$ (AG), AG$^+$VT$^+$ (AGVT) or AG$^-$VT$^+$ (VT) were used for analysis. **(D)** Expression dynamics of the key genes in *GATA3*/*SOX17*/*TFAP2C* clone-derived d6 BT$^+$AG$^+$ cells, and d6ag77 BT$^+$AG$^+$ cells (n = 2, red circles) (see Table S1), overlaid with those in the relevant cell types during the in vitro oogonia/gonocyte differentiation reported in reference 10. **(E)** Scatter-plot comparisons, combined with histogram representations (top and right of scatter plots), of the genome-wide 5 mC levels (genome-wide 2-kb windows) between the indicated cell types. **(F)** Violin-plot representation of the genome-wide 5 mC levels determined by whole-genome bisulfite sequence analysis in the cell types indicated. The mean levels are indicated by red bars. **(G, H)** Heat map representation showing the 5 mC levels in the indicated genomic elements on the autosomes (G) and in the differentially methylated regions of the indicated imprinted genes (H) in the indicated cells. HCP/ICP/LCP, high/intermediate/low-CpG promoters. The color coding is as indicated. **(I)** A model of the transcription factor circuitry driving human primordial germ-cell like cell specification.

# Reagents and tools table

| Reagent/resource | Reference or source | Identifier or catalog number |
|---|---|---|
| **Experimental models** | | |
| BTAG (*BLIMP1-tdTomato* and *TFAP2C-eGFP* knockin reporters in the 585B1 hiPSCs) | Sasaki et al (2015). | N/A |
| BTAG; *SOX17* OE#1 | This study | N/A |
| BTAG; *SOX17* OE#2 | This study | N/A |
| BTAG; *TFAP2C* OE#1 | This study | N/A |
| BTAG; *TFAP2C* OE#3 | This study | N/A |
| BTAG; *SOX17+BLIMP1* OE#9 | This study | N/A |
| BTAG; *SOX17+BLIMP1* OE#15 | This study | N/A |
| BTAG; *TFAP2C+BLIMP1* OE#2 | This study | N/A |
| BTAG; *TFAP2C+BLIMP1* OE#15 | This study | N/A |
| BTAG; *SOX17+TFAP2C* OE#2 | This study | N/A |
| BTAG; *SOX17+TFAP2C* OE#11 | This study | N/A |
| BTAG; *SOX17+TFAP2C* OE#28 | This study | N/A |
| BTAG; *SOX17+TFAP2C+BLIMP1* OE#4 | This study | N/A |
| BTAG; *SOX17+TFAP2C+BLIMP1* OE#9 | This study | N/A |
| BTAG; *SOX17+TFAP2C+BLIMP1* OE#22 | This study | N/A |
| BTAG; *SOX17+TFAP2C+BLIMP1* OE#28 | This study | N/A |
| BTAG; *SOX17+TFAP2C+BLIMP1* OE#31 | This study | N/A |
| BTAG; *SOX17+TFAP2C+MSX2* OE#7 | This study | N/A |
| BTAG; *SOX17+TFAP2C+MSX2* OE#9 | This study | N/A |
| BTAG; *GATA3+SOX17+TFAP2C* OE#1 | This study | N/A |
| BTAG; *GATA3+SOX17+TFAP2C* OE#5 | This study | N/A |
| BTAG; *GATA2+SOX17+TFAP2C* OE#29 | This study | N/A |
| BTAG; *GATA2*$^{-/-}$ #1 | This study | N/A |
| BTAG; *GATA2*$^{-/-}$ #6 | This study | N/A |
| BTAG; *GATA2*$^{-/-}$ #12 | This study | N/A |
| BTAG; *GATA3*$^{-/-}$ #17 | This study | N/A |
| BTAG; *GATA3*$^{-/-}$ #18 | This study | N/A |
| BTAG; *GATA3*$^{-/-}$ #30 | This study | N/A |
| BTAG; *GATA3*$^{-/-}$ #40 | This study | N/A |
| BTAG; *HAND1*$^{-/-}$ #6 | This study | N/A |
| BTAG; *GATA3*$^{-/-}$; *GATA2*$^{+/-}$ #5-1 | This study | N/A |
| BTAG; *GATA3*$^{-/-}$; *GATA2*$^{-/-}$ #5-10 | This study | N/A |
| BTAG; *GATA3*$^{-/-}$; *GATA2*$^{-/-}$; *GATA3* OE #19 | This study | N/A |
| **Recombinant DNA** | | |
| pX335-U6-Chimeric BB-CBh-hSpCas9n (D10A) | Addgene | Cat. no. 42335 |
| **Antibodies** | | |
| Goat anti-SOX17 | R&D Systems | AF1924; RRID: AB_355060 |
| Mouse anti-TFAP2C | Santa Cruz | sc-12762; RRID: AB_667770 |
| Mouse anti-BLIMP1 | R&D Systems | MAB36081; RRID: AB_10718104 |
| Mouse anti-GATA3 | BIOCARE | ACR405A; RRID: AB_10895444 |

**Continued**

| Reagent/resource | Reference or source | Identifier or catalog number |
|---|---|---|
| Rabbit anti-GATA3 | Cell Signaling | CST5852S; RRID:AB_10835690 |
| Rabbit anti-GATA2 | Novus | NBP82581; RRID:AB_11026191 |
| Rabbit anti-GATA2 | Santa Cruz | sc9008; RRID:AB_2294456 |
| Mouse anti-human mitochondria | Merck Millipore | MAB1273; RRID:AB_94052 |
| Goat anti-DDX4 | R&D Systems | AF2030; RRID:AB_2277369 |
| Mouse IgG – HRP conjugated | Sigma-Aldrich | A5906; RRID: AB_258264 |
| Mouse anti-**α** Tubulin | Sigma-Aldrich | T9026; RRID: AB_477593 |
| Mouse IgG – HRP conjugated | Sigma-Aldrich | A5906; RRID: AB_258264 |
| Rabbit IgG – HRP conjugated | Sigma-Aldrich | A6154; RRID: AB_258284 |
| Goat IgG – HRP conjugated | Sigma-Aldrich | A5420; RRID: AB_258242 |
| SSEA1 (CD15) microbeads for human and mouse | Miltenyi Biotec | 130-094+530 |
| CD31 microbeads for mouse | Miltenyi Biotec | 130-097-418 |
| Oligonucleotides and sequence-based reagents | | |
| qRT-PCR primers | This study | Table S3 |
| Chemicals, enzymes and other reagents | | |
| StemFit AK03N | Ajinomoto | N/A |
| iMatrix-511 | Nippi | |
| Puromycin | Thermo Fisher Scientific | A1113803 |
| G418, Geneticin | Thermo Fisher Scientific | #10131035 |
| Hygromycin B | Thermo Fisher Scientific | #10131035 |
| Doxycycline | Takara-Clontech | Z1311N |
| Fibronectin | Millipore | FC010 |
| GMEM | Thermo Fisher Scientific | #11710035 |
| Knockout serum replacement | Thermo Fisher Scientific | A3181502 |
| Activin A | Peprotech | 120-14E |
| CHIR99021 | TOCRIS | #4423 |
| Y27632 | FujiFilm | 030-24021 |
| BMP4 | R&D Systems | 314-BP |
| SCF | R&D Systems | 255-SC |
| EGF | R&D Systems | 236-EG |
| LIF | Millipore | LIF1010 |
| LDN193189 | StemGent | 04-0074 |
| Glutamax | Thermo Fisher Scientific | 35050-061 |
| HEPES | Thermo Fisher Scientific | 15630-106 |
| **α**-Minimum Essential Medium | Thermo Fisher Scientific | 32571-036 |
| L-ascorbic acid | Sigma-Aldrich | A4403 |
| Software | | |
| FACSDiva Software | BD Biosciences | N/A |
| DAVID (v6.8; GO analysis) | https://david.ncifcrf.gov/ | N/A |
| FV10-ASW | Olympus | N/A |
| R (v3.6.0; PCA, DEG, and graphs) | https://www.R-project.org | N/A |
| Bowtie2 v2.2.7 | http://bowtie-bio.sourceforge.net/bowtie2/index.shtml | N/A |
| TopHat v2.1.0 | https://ccb.jhu.edu/software/tophat/index.shtml | N/A |

| Reagent/resource | Reference or source | Identifier or catalog number |
|---|---|---|
| HTSeq v0.9.1 | https://htseq.readthedocs.io/en/master/overview.html | N/A |
| ImageJ/Fiji | Fiji.sc | N/A |
| Trim_galore v0.6.3 | https://www.bioinformatics.babraham.ac.uk/projects/trim_galore/ | N/A |
| cutadapt v118 | http://cutadapt.readthedocs.io/en/stable/guide.html | N/A |
| Bismark v0.22.1 | https://www.bioinformatics.babraham.ac.uk/projects/bismark/ | N/A |
| SAMtools v1.9 | http://samtools.source-forge.net | N/A |
| Other | | |
| pGEM-T Easy Kit | Promega | A3600 |
| Gateway LR Clonase Enzyme Mix | Thermo Fisher Scientific | #11791043 |
| v-bottom 96-well plate | Greiner | #651970 |
| RNeasy Micro Kit | QIAGEN | #74004 |
| NucleoSpin RNA XS | Macherey-Nagel | #740902 |
| Qubit RNA HS assay kit | Thermo Fisher Scientific | Q32855 |
| PowerSYBR Green PCR Master Mix | Thermo Fisher Scientific | #4367659 |
| Qubit dsDNA HS assay kit | Thermo Fisher Scientific | Q32851 |
| Protein Quantification Assay | Macherey-Nagel | #740967 |
| ECL Western Blotting Detection Reagent | GE Healthcare Life Sciences | RPN2106 |
| EZ DNA Methylation-Gold Kit | Zymogen | D5005 |
| DP72 | Olympus | N/A |
| FV1000-IX81 confocal microscope system | Olympus | N/A |
| CFX384 Touch Real-Time PCR detection system | Bio-Rad Laboratories | N/A |
| NextSeq500/550 | Illumina | N/A |
| Hiseq2500 | Illumina | N/A |

10–15 min at 37°C with gentle pipetting every 5 min. Trypsin was neutralized with a 5× volume of 10% FBS in DMEM, and the resuspended cells were processed with FACS Aria III system (BD Biosciences) and analyzed with FACS Diva software.

The method for selecting CRISPR-mediated knockout clones is described in the section "Generation of knockout lines."

### cDNA amplification, qRT-PCR and RNA-seq analysis

Total RNA was extracted from the frozen cell pellets using RNeasy kits (QIAGEN) or NucleoSpin RNA kits (Macherey-Nagel) following the manufacturers' instructions. The amount of RNA was measured with Qubit 2.0 (Thermo Fisher Scientific) and the cDNAs were synthesized through amplification of their 3′ ends starting from 1 ng of total RNA as described previously (29). The RNA sample was mixed with ERCC (External RNA Controls Consortium; Thermo Fisher Scientific) spike RNA and then reverse transcribed with V1-(dT)24 primer using SuperScript III for 5 min at 50°C. SuperScript III was immediately inactivated at 70°C for 10 min, and the excess primer was digested with Exonuclease I (TakaraBio) for 30 min at 37°C followed by heat inactivation for 25 min at 80°C. Then the poly A tail was added to the cDNA with Terminal Deoxynucleotidyl Transferase

(TakaraBio) for 15 min at 37°C and heat inactivated for 10 min at 70°C. Subsequently, PCR amplification was done using ExTaq HS polymerase (TakaraBio); the first cycle was run with V3-(dT)24 primer alone, followed by 14 cycles using both V1-(dT)24 and V3-(dT24). The PCR product was then purified and the primer dimers were removed by adding a 0.6× volume of AMPure XP (Agencourt), washed with 80% ethanol two times, and eluted with 50 $\mu$l 5 mM Tris–HCl (pH 8.0) on a magnetic stand. In some cases, an AxyPrep MAG PCR Clean Up Kit (Corning) was used in place of AMPure XP; the two provided comparable results.

qRT-PCR was performed with PowerSYBR Green PCR Master Mix (Thermo Fisher Scientific) on a CFX384 Real-Time PCR Detection System (Bio-Rad) using the primers listed in Table S3. The quality of the amplified cDNA was assessed according to the Ct values by qPCR of the ERCC spike RNA and the housekeeping genes (*PPIA* and *RPLP0*).

The cDNA library was prepared as described previously (78). 5 ng aliquots of quality-controlled cDNA samples were further amplified by PCR using ExTaq HS (TakaraBio) with the N-V3 (dT)24 and V1 (dT)24 primers for four cycles, purified with three rounds of binding, washing and eluting steps with AMPureXP, and then fragmented with a Covaris E220 sonicator. The fragmented products were end-polished with T4

DNA polymerase (NEB) and T4 polynucleotide kinase (NEB) for 30 min at 20°C. The products were then purified again with a 0.7× volume of AMPureXP, followed by addition of a 0.9× volume of AMPureXP to the supernatant, and a final washing and elution. To attach adaptor sequences, the cDNA solution was treated first with Rd2SP-V1(dT)20 primer using ExTaqHS, followed by addition of Rd1SP-adaptor with T4 DNA ligase (NEB), and purified with a 0.8× volume of AMPureXP. The adapter attached cDNA was then PCR amplified using Nextera XT Index 1 (N7XX) and Index 2 (S5XX) Primers (Illumina) with ExTaqHS for 10 cycles and purified by two washings with a 0.9× volume of AMPureXP.

The quality and quantity of the resultant library DNAs were evaluated by the LabChip GX (Perkin Elmer), the Qubit dsDNA HS assay kit, and the Taqman-qPCR assay using Thunderbird Probe qPCR mix (TOYOBO) and a TaqMan probe (Ac04364396; Applied Biosystems). The sequence data were acquired using NextSeq 500 (Illumina). Conversion of the sequence read data into expression levels was performed as described previously (29, 78). The reads were first processed with cutadapt-1.3 (79) to trim the V1 and V3 adaptor sequences and poly-A sequences. The trimmed reads longer than 30 bp were then mapped onto the GRCh38.p2 genome using Tophat v2.1.0/Bowtie2 v2.2.7, with the "-no-coverage-search" option (80). The expression levels (reads per million-mapped reads: RPM) were calculated from these mapped reads using the HTSeq v0.9.1 with default settings, and the GRCh38.p2 reference gene annotations were modified, where necessary, so that the transcript termination sites were extended up to 10 kb downstream.

### Data analysis of the RNA-seq

All statistical analyses were performed on R (ver 3.6.0) with stringr, gplots, and prcomp packages. The expression data were first converted into $\log_2(RPM + 1)$ values, the genes with maximum $\log_2(RPM + 1) < 4$ (equivalent to 10–20 copies per cell) in all the samples were excluded, and the distribution of the expression levels was assessed with boxplots to confirm the quality of the samples. For hPGCLC induction, 180 samples were added to the 230 samples used in the former study with 12,909 genes (13), and for xenogeneic reconstituted ovary series, three iMeLC samples and three ag77 samples were newly collected and analyzed with 30 samples collected in the previous study (10) with 12,737 genes.

All the heat maps were drawn with the heatmap.2() function, the correlation coefficient between BMP-induced and overexpressed samples was calculated with the cor() function with "Pearson correlation," and the PCA was performed with the prcomp() function. DEGs among SOX17/TFAP2C/BLIMP1 samples were defined as $P < 0.01$ by one-way ANOVA among all time points and conditions of SOX17/TFAP2C/BLIMP1 series, $P < 0.01$ in Tukey–Kramer post hoc test for multiple comparisons, $\log_2(RPM + 1) > 4$ in the higher expression group, and more than onefold change of the mean expression values. GO analysis was performed on the DAVID website (https://david.ncifcrf.gov). The GO terms were extracted from the Biological Process (GOTERM_BP_DIRECT) and the pathway terms from KEGG_PATHWAY (Kyoto Encyclopedia of Genes and Genomes) (http://www.genome.jp/kegg/) (81).

Single-cell transcriptome data of cynomolgus monkey embryos was retrieved from the Gene Expression Omnibus database under accession numbers GSE67259, GSE74767 and GSE76267 (30, 31). The following cells were extracted: post-implantation parietal trophectoderm (designated Post-paTE), post-implantation early epiblast (PostE-EPI), post-implantation late epiblast (PostL-EPI), gastrulating cells group 1 (G1), gastrulating cells group 2a (G2a), gastrulating cells group 2b (G2b), visceral endoderm or yolk sac endoderm (VE/YE), extraembryonic mesenchyme (EXMC), early PGCs (ePGC), and late PGCs (lPGC). The heat maps were drawn with the heatmap.2() function on R (ver 3.6.0). Note that SOX17 and GATA2 were named LOC101925698 and LOC101865311, respectively, in these datasets.

### Immunofluorescence of iMeLC aggregates and cynomolgus monkey embryos

Cynomolgus monkey embryos were isolated as described previously (18, 30). The samples were formalin-fixed, paraffin-embedded, and stored at 4°C. The sectioned samples were first deparaffinized and hydrated followed by antigen retrieval with HistoVT One (Nacalai) at 90°C for 35 min. After washing in PBS, the samples were permeabilized and blocked with blocking solution (5% donkey serum, 0.2% Tween 20, in PBS) for 2 h at ambient temperature. The samples were then treated with primary antibodies in the blocking solution overnight at 4°C, washed three times with PBS, incubated with secondary antibodies at room temperature for 1 h, washed three times with PBS, mounted in VECTASHIELD Mounting Medium with DAPI (Vector Laboratories), and imaged under an Olympus FV1000 confocal microscope.

For the immunofluorescence analysis of reconstituted xenogeneic ovaries, the harvested reaggregates were fixed with 4% paraformaldehyde at 4°C for 2 h, followed by two-step cryoprotection with 10% and 30% sucrose dissolved in PBS at 4°C for 1 h and overnight, respectively, and finally, freezing in Optimal Cutting Temperature (OCT) compound (Sakura Finetek). The sectioned samples were washed with PBS to remove the OCT compound, and then permeabilized and blocked using the protocol described above.

### Xenogeneic reconstituted ovary (xrOvaries) culture

d6 hPGCLCs were further differentiated by aggregation with mouse female gonadal somatic cells at embryonic day (E) 12.5, which we termed xrOvaries as described previously (10, 11). Pregnant ICR females were euthanized by cervical dislocation and the E12.5 embryos were dissected in DMEM containing 10% FBS (Hyclone), 2 mM GlutaMax, 10 mM HEPES, and 100 U/ml penicillin/streptomycin. Fetal ovaries were dissected out with tungsten needles and dissociated into single cells with 0.25% Trypsin, the endogenous mouse PGCs were removed by MACS with anti-CD31 and anti-SSEA1 antibodies (Miltenyi), and the remaining fetal ovarian somatic cells were aliquoted and frozen until use. Thawed somatic cells (75,000 cells/well) were mixed with d6 hPGCLCs (5,000 cells/well) and cultured in a Lipidure-coated U-bottom 96-well plate with GK15 medium containing 10 μM Y27632 (Tocris) to form floating cellular aggregates. After 2 d under this condition, the xrOvaries were transferred onto Transwell-COL membrane inserts (Corning) with a mouth pipette and maintained as an air-liquid interface

culture in α-MEM with 10% FBS, 55 μM 2-mercaptoethanol, 150 μM l-ascorbic acid (Sigma-Aldrich), and 100 U/ml penicillin/streptomycin. The xrOvaries were cultured at 37°C, 5%CO₂ in air and half of the medium was replaced every 3 d until harvest. All the reagents were from Thermo Fisher Scientific unless otherwise specified.

### WGBS

Genomic DNA was prepared from 5,000 cells, then resuspended in lysis buffer containing 0.1% SDS and 1 μg/μl Proteinase K in DNase Free Water (GIBCO) and incubated at 37°C for 60 min; the Proteinase K was then heat inactivated at 98°C for 15 min. This lysate was spiked with 150 pg of unmethylated λ phage DNA (Promega), based on the estimation that the amount of genomic DNA was 6 pg per cell and the phage DNA accounted for 0.5% of the input genomic DNA. Bisulfite conversion and library construction were performed by the post-bisulfite adaptor tagging (PBAT) method (82). The detailed protocol of the PBAT method has been published and is freely available at the website of the International Human Epigenome Consortium (http://www.crest-ihec.jp/english/epigenome/index.html). All the reagents used were the same as described in the protocol, except that Phusion Hot Start II DNA Polymerase (Thermo Fisher Scientific) and an AxyPrep MAG PCR Clean-Up Kit (Corning) were substituted for the Phusion Hot Start High-Fidelity DNA polymerase and Agencourt AMPure XP, respectively. Deep sequencing was performed on an Illumina Hiseq 2500 system to generate 101-nucleotide single-end sequence reads, and cluster generation and sequencing were implemented in single-read mode using a TruSeq SR Cluster Kit v3-cBot-HS and TruSeq SBS Kit v3-HS (Illumina) following the manufacturer's instructions.

### WGBS data processing

The WGBS data were first processed with Trim_Galore v0.6.3 (https://www.bioinformatics.babraham.ac.uk/projects/trim_galore/)/cutadapt v1.18 (http://cutadapt.readthedocs.io/en/stable/guide.html) with the "--clip_R1 4," "--trim1" and "-a AGATCGGAAGAGC" options to trim away low-quality bases (four bases from the 5′ ends, one base from the 3′ends and bases with quality score <20) and the adapter sequences. The qualified reads were then mapped onto the human genome (GRCh38.p2) using Bismark v0.22.1 with the "--pbat" option, and the cytosine/methyl-cytosine count at every CpG site on the genome was determined by the bismark_methylation_extractor program included in the package.

For the genome-wide analysis, all the CpG sites with read depth ≥4 were used for the following analyses. The percent methylations of individual CpG sites were plotted in the Violin plot using the vioplot package. For the scatterplots, the average percent methylations of the CpG sites in 2-kb non-overlapping bins that carried four or more CpG sites were used. The scatterplots were overlaid with contour plots to enhance the visibility of the plot density, and a histogram in each scatterplot shows the frequency of CpG methylation at 5% intervals within the samples.

To compare the ratio of DNA methylation in particular regions, the genomic DNA sequence was grouped into the following regions. Promoters were defined as the region between 900-bp upstream and 400-bp downstream of the transcription start sites, and the promoters with high CpG (HCP), intermediate CpG (ICP), and low CpG (LCP) were classified according to the previous report (83). Data for the CpG islands (84) and human imprint loci (85) were downloaded from public repositories and converted to GRCh38 format with LiftOver. No statistical analysis was performed on WGBS data.

## Data and Code Availability

All the sequencing data have been deposited in the Gene Expression Omnibus database under accession number GSE154691 (RNA-seq: GSE154688; WGBS: GSE154690) and the R script is available on request.

## Supplementary Information

## Acknowledgements

We thank the members of our laboratory for their helpful input on this study. We are grateful to Y Nagai, N Konishi, E Tsutsumi, and M Kawasaki of the Saitou Laboratory, and J Asahira and M Kabata of the Yamamoto Laboratory for their technical assistance. This work was supported in part by Grants-in-Aid for Scientific Research from Japan Society for the Promotion of Science (JSPS) (15K08267 and 20K06654) to Y Kojima and by a Grant-in-Aid for Specially Promoted Research from JSPS (17H06098), an Exploratory Research for Advanced Technology Grant from the Japan Science and Technology Agency (JST-ERATO) (JPMJER1104), a Grant from Human Science Frontier Program (RGP0057/2018), and grants from the Pythias Fund and Open Philanthropy Project to M Saitou.

### Author Contributions

Y Kojima: conceptualization, data curation, formal analysis, supervision, funding acquisition, validation, investigation, visualization, methodology, project administration, and writing—original draft, review, and editing.
C Yamashiro: resources, investigation, and methodology.
Y Murase: resources, investigation, and methodology.
Y Yabuta: resources, formal analysis, investigation, and methodology.
I Okamoto: resources, investigation, and methodology.
C Iwatani: resources, investigation, and methodology.
H Tsuchiya: resources, investigation, and methodology.
M Nakaya: resources, investigation, and methodology.
T Tsukiyama: resources, investigation, and methodology.
T Nakamura: resources, investigation, and methodology.
T Yamamoto: resources, investigation, and methodology.
M Saitou: conceptualization, formal analysis, supervision, funding acquisition, investigation, methodology, project administration, and writing—original draft, review, and editing.

## Conflict of Interest Statement

The authors declare that they have no conflict of interest.

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
