## [Reviewer comments · Life Science Alliance]

Life Science Alliance

GATA transcription factors, SOX17 and TFAP2C drive the human germ-cell specification program

Mitunori Saitou, Yoji Kojima, Chika Yamashiro, Yusuke Murase, Yukihiro Yabuta, Ikuhiro Okamoto, Chizuru Iwatani, Hideaki Tsuchiya, Masataka Nakaya, Tomoyuki Tsukiyama, Tomonori Nakamura, and Takuya Yamamoto

DOI: <https://doi.org/10.26508/lsa.202000974>

Corresponding author(s): Mitunori Saitou, Institute for the Advanced Study of Human Biology, Kyoto University and Yoji Kojima, Institute for the Advanced Study of Human Biology, Kyoto University

Review Timeline:

Submission Date:	2020-11-27
Editorial Decision:	2020-11-30
Revision Received:	2021-01-03
Editorial Decision:	2021-01-06
Revision Received:	2021-01-07
Accepted:	2021-02-05

Scientific Editor: Shachi Bhatt

Transaction Report:

Please note that the manuscript was previously reviewed at another journal and the reports were taken into account in the decision-making process at Life Science Alliance.

November 30, 2020

Re: Life Science Alliance manuscript #LSA-2020-00974-T

Prof. Mitinori Saitou
Graduate School of Medicine, Kyoto University
Department of Anatomy and Cell Biology
Yoshida-Konoe-cho, Sakyo-ku
Kyoto 606-8501
Japan

Dear Dr. Saitou,

Thank you for submitting your manuscript entitled "GATA transcription factors interlink the SOX17 and TFAP2C pathways to generate the human germ-cell specification program" to Life Science Alliance (LSA).

For a brief overview, this manuscript was previously submitted and reviewed at an alliance journal, but was rejected post-review due to concerns raised by reviewers. The manuscript and the accompanying reviews were then discussed with LSA, and the study was deemed to be appropriate for LSA provided the authors addressed the following,

- + all the points raised by Reviewer 1 are addressed through discussion. No further experiments will be required
- + all the minor points raised by Reviewer 2 and all the points raised by Reviewer 3 are addressed
- + the writing and figures are significantly revised to improve readers' understanding and flow
- + the citations are improved (as pointed out by Rev 1), and the recent PNAS paper from Dr. Clark's group is discussed (as pointed out by Rev 3)

We encourage the authors to address the above-mentioned points and submit a revised manuscript back to LSA, along with a point-by-point rebuttal to the reviewers' comments raised at the previous journal.

Thank you for this interesting contribution to Life Science Alliance. We are looking forward to receiving your revised manuscript.

Sincerely,

Shachi Bhatt, Ph.D.
Executive Editor
Life Science Alliance
<https://www.lsa-journal.org/>
Tweet @SciBhatt @LSAJournal

- A letter addressing the reviewers' comments point by point.
- An editable version of the final text (.DOC or .DOCX) is needed for copyediting (no PDFs).
- High-resolution figure, supplementary figure and video files uploaded as individual files: See our detailed guidelines for preparing your production-ready images, <https://www.life-science-alliance.org/authors>
- Summary blurb (enter in submission system): A short text summarizing in a single sentence the study (max. 200 characters including spaces). This text is used in conjunction with the titles of papers, hence should be informative and complementary to the title and running title. It should describe the context and significance of the findings for a general readership; it should be written in the present tense and refer to the work in the third person. Author names should not be mentioned.

B. MANUSCRIPT ORGANIZATION AND FORMATTING:

We would like to sincerely thank the referees for their constructive comments, which we have used as the basis for revising our manuscript.

Referees' Comments:

Referee #1:

The manuscript by Kojima et al. used human embryonic stem cells (hESC) differentiation model to define transcription factors that are essential to induce germ-cell fate in pluripotent precursors. Using overexpression models, loss-of-function approaches as well as rescue of function approach, authors concluded that GATA family transcription factors, GATA3 and GATA2, in coordination with SOX17 and TFAP2C, instigate differentiation to human primordial germ cell-like cells (hPGCLC). The in vivo relevance of GATA, SOX17 and TFAP2C transcriptional axis were supported with the findings that GATA3, SOX17 and TFAP2C co-express in early primordial germ cells (PGCs) of a developing primate embryo. Furthermore, authors tested developmental potential of GATA3/SOX17 and TFAP2C-induced hPGCLCs into oogenia/gonocytes in via xenogenic embryo reconstitution assay. Overall it is an interesting study that implicates importance of GATA3/GATA2 (more specifically GATA3) in PGC development.

Response 1. We would like to thank the Referee for the encouraging comments on our manuscript.

Tremendous amount of experiments were performed. However, the conclusion of the study is not supported by experimental outcomes and several aspects of the study raise significant concerns, which are mentioned below.

1. Authors considered that the any combination of transcription factor overexpression would lead to the similar ground state of incipient mesoderm like cells (iMeLCs) and never tested whether different combination of TF overexpression leads to an altered gene expression patterns in iMeLCs. It is possible that the impaired hPGCLC differentiation is due to defect in gene expression program that are instigated prior to induction of hPGCLC specific differentiation.

Response 2. We suspect that this comment arose from a misunderstanding by the Referee. iMeLCs were induced from hiPSCs by activin A and CHIR99021 (a WNT signal activator), and the transgenes were induced by Dox in iMeLCs aggregates (i.e., after iMeLC differentiation) in order to explore whether they are sufficient to differentiate iMeLCs into hPGCLCs. Thus, the transgenes were not overexpressed during the process of hiPSCs-to-iMeLC induction.

A leakage of the transgenes in the absence of Dox might affect the properties of iMeLCs. However, as shown in Fig 2A–2E (in both the original and revised manuscript), iMeLCs induced by activin A and CHIR99021 from parental hiPSCs or hiPSCs bearing the transgenes (no Dox addition) showed very similar transcriptomes, and as shown in Fig

1E, iMeLCs bearing any combination of the transgenes differentiate into BT⁺AG⁺ hPGCLCs in response to BMP4 in a reproducible manner, excluding the possibility that the properties of iMeLCs themselves are compromised by the transgenes.

We stressed that iMeLCs were induced by activin A and CHIR99021 in the revised manuscript (the “**SOX17, TFAP2C, and BLIMP1 are not sufficient to generate hPGCLCs**” section, the fourth paragraph in the revised manuscript).

2. From Figure 1E, it appears that ectopic expressions of either SOX17 (S) or SOX17 and TFAP2C (SA) or TFAP2C and BLIMP1 (AB) affects (prominent differences in cell morphology, size etc.) BMP4-dependent hPGCLC differentiation program. It is not understood why such an effect is observed if ectopic expression levels are similar to BMP4-induced expressions from the endogenous loci (As claimed in Fig. 1C).

Is GATA3 or GATA2 expression induced in these cells upon BMP4 treatment?

It is also not clear why only SA clones show 90% BT⁺/AG⁺ cells upon BMP4 treatment. All of these data are confusing and indicates various transcriptional outcomes in different TF-combinations. Authors should explain these outcomes.

The claim that BLIMP1 represses its own transcription (mentioned in para 2, page 7) is not clear from experimental data. Also, Fig. EV1B should included samples from d2 PGCLC to show relative protein levels.

Response 3. We thank the Referee for these comments. First, please note that the photographs shown in Fig 1E are those for iMeLC aggregates (~5,000 cells per aggregates initially) under the indicated transgene-expression conditions, and therefore show the size and morphology of the aggregates, but not the cells themselves.

The reason why the sizes of the iMeLC aggregates of the *SOX17*, *SOX17/TFAP2C*, and *TFAP2C/BLIMP1* clones became smaller when we treated the cells with both BMP4 and Dox is unclear, although there is a possibility that a subtle but significant difference, e.g., > 2-fold, of the expression levels of *SOX17* or *TFAP2C* or *BLIMP1* could cause such an effect. For example, *Blimp1/BLIMP1* is known to induce cell-cycle arrest in various contexts (see (1, 2) for review).

In Fig 2H of the original manuscript (the revised manuscript as well), we showed that *GATA3* is expressed at a high level in response to BMP4 in the iMeLC aggregates of the *SOX17/TFAP2C/BLIMP1* clones. We evaluated the transcriptome data, which indicated that the iMeLC aggregates of the *SOX17/TFAP2C* clones or the *SOX17/TFAP2C/BLIMP1* clones treated with BMP4 and Dox express both *GATA3* and *GATA2* (Fig S3A in the revised manuscript). Thus, we reason that *GATA3* and *GATA2* are induced by BMP4 in transgene-expressing clones.

As a potential reason why the iMeLC aggregates of the *SOX17/TFAP2C* clones showed ~90% BT⁺AG⁺ cells upon BMP4 treatment, we speculate that *SOX17* and *TFAP2C* expression is a rate-limiting, although not sufficient, event for hPGCLC specification, and therefore the Dox-induced expression of *SOX17* and *TFAP2C* creates a state highly competent for BMP-induced hPGCLC specification.

The reason why we claimed that “*BLIMP1* appears to have an activity to repress itself” is that when we compared the induction levels of *BLIMP1-tdTomato* (BT) between the iMeLC aggregates of the *SOX17* or *SOX17/BLIMP1* clones, and between the iMeLC aggregates of the *SOX17/TFAP2C* or *SOX17/TFAP2C/BLIMP1* clones, the clones additively expressing *BLIMP1* showed lower BT induction levels (Please compare the BT levels in the third rows (labeled D) between the S#1 and SB#9 panels, and between the SA#2 and SA#4 panels in Fig 1E in the original manuscript; the revised manuscript as well).

To make the first section of the **RESULTS** more succinct, we decided to remove the statement that “*BLIMP1* appears to have an activity to repress itself” in the revised manuscript, as this statement is not necessarily critical in this manuscript. In regard to the other points, we added the relevant data and discussion in a succinct manner in the revised manuscript (the “*SOX17, TFAP2C, and BLIMP1 are not sufficient to generate hPGCLCs*” section, Fig 2H and S3A in the revised manuscript).

3. *GATA2 expression is not detected in post-implantation primate embryos and from studies on other mammalian system it is clear that in vivo GATA2 is only expressed in the TE lineage. Thus, it seems that the role of GATA2 in PGC development is a stretch and could be an artifact of the in vitro experimental system. This raises several questions. (i) Is GATA2 induced in GATA3-KO cells? (ii) Are GATA3 and GATA2 expressions detected in a mutually exclusive fashion in single-cell transcriptome analyses with cyPGCs?*

Response 4. Please note that in the original manuscript (the revised manuscript as well), we provided data in Fig 2J showing that *GATA2* is detectable in at least 6 out of 16 single-cell cDNAs from cynomolgus monkey early PGCs, and *GATA3* and *GATA2* show expression in an overlapping manner. Furthermore, in Fig 6E, we showed that in addition to PGCs, *GATA2* is expressed in gastrulating cells, extraembryonic mesenchyme, and post-implantation late epiblasts, as well as in post-implantation parietal TEs. Therefore, *GATA2* is clearly expressed in PGCs and other cells in post-implantation cynomolgus monkey embryos.

Also, we provided data showing that *GATA2* and *GATA3* are normally induced in *GATA3*^{-/-} and *GATA2*^{-/-} cells, respectively, in Fig 5B.

4. *In paragraph one of page 12 it is claimed that due to reduced differentiation efficiency of GATA3-KO cells, compared to Control and GATA2-KO cells prompted*

authors to hypothesize that GATA3 and GATA2 have a compensatory function. It is surprising and none of the experiments test this concept. Authors rescued only GATA3 expression in double KO cells. It is important to rescue both GATA2 and GATA3 in double KO cells to show that either of them are sufficient to induce PGC differentiation program.

Response 5. Please note that we provided data showing that *GATA2*, *SOX17* and *TFAP2C* expression induces BT⁺AG⁺ hPGCLCs in Figs 3E, 3G, and EV3F, and discussed the results in the fourth paragraph of the “**GATA TFs, SOX17, and TFAP2C drive hPGCLC induction**” section in the original manuscript (and in the corresponding figures and texts in the revised manuscript as well). Together, these findings demonstrate the compensatory function of *GATA3* and *GATA2*, and show that either of *GATA3* or *GATA2* is sufficient to induce the PGC differentiation program.

It is also important to show that they have a common gene regulatory program (such as common target genes) during hPGCLC specification.

Response 6. This is an important point that we would like to explore carefully in a future study, as it is beyond the scope of the present manuscript.

5. Overall, the only definitive conclusion from this study is an important but non-essential (at least in vitro) role of GATA3 during germ cell fate specification. None of the data shows that regulatory pathways via GATA, SOX17 and TPAP2C converge during hPGCLC specification. Thus, it is not clear what authors meant by the claim that "GATA3 and GATA2 interlink SOX17 and TFAP2C pathways to generate hPGCLC". There is no mechanistic study or direct evidence to support this claim.

Response 7. We made the claim based on the functional evidence: Our data show that *SOX17* alone does not induce endogenous *SOX17* or *TFAP2C*, *TFAP2C* alone does not induce endogenous *SOX17* or *TFAP2C*, and furthermore, the *SOX17* and *TFAP2C* combination does not induce endogenous *SOX17* or *TFAP2C* (Figs 1C, 1E, 1G in the original manuscript), indicating that the pathways regulated by *SOX17* and *TFAP2C* are independent and do not activate each other.

On the other hand, when *GATA3* or *GATA2* are co-expressed with *SOX17* and *TFAP2C*, both endogenous *SOX17* and *TFAP2C* are activated, and the overall hPGCLC specification program commences (Figs 3 and 4A in the original manuscript). We consider that these functional data serve as evidence for our claim that “GATA transcription factors interlink the *SOX17* and *TFAP2C* pathways to generate hPGCLCs”, although the underlying mechanism remains to be elucidated.

In response to the Referee’s comment, however, we revised the title of the manuscript to “**GATA transcription factors, SOX17 and TFAP2C drive the human germ-cell specification program**”, which conveys our findings in a more straightforward manner.

Minor comment:

(1) Authors should be more respectful to prior studies showing complementary role of GATA2 and GATA3 in other cellular contexts, such as during TE development. It seems authors mainly cited a review paper (Tremblay et al., 2018) rather than actual reports.

Response 8. Please note that in the fourth paragraph of the **DISCUSSION** section in the original manuscript, we discussed the point raised by the Referee with appropriate references, including those showing a complementary role of *GATA2* and *GATA3* in other cellular contexts, such as during TE development.

The reason why we cited review papers in the first paragraph of page 9 is that we sought in this paragraph to introduce what is known about *GATA3* and *MSX2* in a concise manner. We evaluated the compensatory role of *GATA3* and *GATA2* for hPGCLC specification in later sections and discussed it in the **DISCUSSION**.

In response to the Referee's comment, we added the phrase "see **DISCUSSION** for the roles of *GATA3* in relevant contexts" along with additional references in the relevant sections in the revised manuscript (the "**GATA TFs, SOX17, and TFAP2C drive hPGCLC induction**" section, the first and fourth paragraphs in the revised manuscript).

(2) Authors should include actual reference instead of "ref" at the end of line 4 of paragraph 2 in page 12.

Response 9. We thank the Referee for pointing out this error. We provided an appropriate reference in the revised manuscript (the "**Critical requirements of the GATA TF paralogs for hPGCLC specification**" section, the third paragraph in the revised manuscript).

Referee #2:

The manuscript entitled "GATA transcription factors interlink the SOX17 and TFAP2C pathways to generate the human germ cell specification program" describes the importance of GATA2 & 3 factors for the in vitro induction of human PGCLs (Precursors of Germ Cell like).

The authors have established in vitro reporter systems of human induced pluripotent stem cells whereby they examine the requirement for SOX17/TFAP2C/BLIMP1 to confer germ cell fate, and conclude that these factors are not sufficient for hPGCLCs induction. Therefore, the authors have performed in silico search of previously published data

(from the Saitou group in hPGCLCs transcriptome and single cell RNA seq data from cynomolgus monkey PGCs) in order to identified independent TFs for hPGCLC derivation. This search direct them on GATA TFs (here focus is on MSX2 or GATA 2 or 3), which they went on to test if their over expression or dose reduction are important to drive hPGCLC induction.

They also analyse expression pattern of the transcription factors they describe in their study to be critical for PGC induction using immunofluorescent approaches on paraffin sections of cynomolgus monkey embryos. These approaches are important to better dissect inter-species difference for germline formation, in particular to understand different with common mouse model used for functional testing.

Altogether, their data lead to the conclusion that GATA2&3 are BMP effectors to drive the germ cell like specification program. This work introduces GATA2/3 players as part of the core component pathways driving germline cell identity and aims at scientists dissecting this pathway using this cell system. These conclusions are in direct agreement with their previous work and the conclusions are supported by the experiments. However, the insights are moderate and seems to be relevant to a rather specialized audience.

Comments

Overall, the manuscript and the figures are not easy to grasp. The manuscript could gain in quality to highlight better the results and less the methods. The use of too many acronyms makes the text very difficult to read. It also limits its reading by non-specialist of the field. Clarity should not be left for brevity.

Response 1. We thank the Referee for this suggestion. In response to the Referee's comments, we removed abbreviations/acronyms as much as possible, except those that are generally used, and revised the manuscript and figures so that they better highlight the results and flow more easily. We sincerely hope that the revised manuscript meets with the Referee's approval.

Most of the graphs of the figure do not contain y axis label rendering the reading unpleasant.

Response 2. We re-examined the graphs in the figures of the original manuscript and found that essential information regarding what the y axes represent was provided in all the graphs, and further information was provided in the legends to figures.

It may be that the Referee found it difficult to evaluate the graphs of the original manuscript because they contained many abbreviations/acronyms, as pointed out above, and because some of the graphs, e.g., Fig 1C, may have been unnecessarily complicated. We therefore removed such abbreviations/acronyms wherever possible and revised the figures to make it more clear what is depicted. We very much hope that the revised figures meet with the Referee's approval.

The authors should discuss at least discuss the role of Zglp1 in their human cell system, in the context of the BMP cascade and their study.

Response 3. Please note that *Zglp1* is a factor that functions in mice when PGCs that complete epigenetic reprogramming in embryonic ovaries (oogonia) differentiate into oocytes, but not when the epiblasts differentiate into PGCs (3). Therefore, the context is clearly different from what we examined in the present manuscript, i.e., the mechanism of human PGC specification. We discussed the differences of the mechanisms between mouse and human PGC specification and the broad implications of our study in the first and seventh paragraph, respectively, of the **DISCUSSION** in the original/revised manuscript.

Referee #3:

The authors used in vitro hPGCLC system to probe the necessary/sufficiency role of transcription factors sox17, tfap2c and blimp1 in inducing the transcriptional programme of PGCLCs. They do this by performing several overexpression of such transcription factors, coupled by gene reporters and RNAseq. They identify GATA and MSX factors as highly expressed and further test the role of expressing ectopically GATA2 and GATA3 in driving the transcriptional programme (and to a lesser extent the DNA methylation programme) of PGCLCs. They also interrogate the expression of BMP-downstream targets in cynomolgus monkeys.

The manuscript is extremely difficult to read, and the data in the figures is also very difficult to comprehend. The authors have a disproportionate use of abbreviations that do not really reflect the biology behind (e.g. starting with S, SB, BT, AG, BTAG etc throughout), which are more a lab 'slang' and make the data and the reading very difficult to access.

I urge the authors to make a big effort in rendering their manuscript understandable and accessible, as well as the figures too. I spent far too much time in trying to understand their abbreviations, and the flow of the figures. The graphs are cramped, the labels are insufficient and the abbreviations are not meaningful in terms of biological terms.

Response 1. In response to the Referee's comments, we removed abbreviations/acronyms as much as possible, except those that are generally used, and revised the manuscript and the figures so that they better highlight the results and flow more easily. We sincerely hope that the revised manuscript meets with the Referee's approval.

Other comments:

Figure 1 - it is unclear what genes were used as internal control to calculate gene

expression changes in panel C.

Response 2. Please note that in the original manuscript, we clearly stated in the legends to Fig 1C that “ Δ Ct was calculated from the average Ct value of two housekeeping genes, *RPLP0* and *PPIA* (set as 0)”. In response to the Referee’s comment, we provided this information in the figure panels as well in the revised manuscript.

Figure 1 - panel C is extremely hard to follow, and is cramped - the key to the colours is also very difficult to read. Please correct the layout of the data, perhaps doing a heatmap of changes in gene expression may be a better option to present these dataset.

Response 3. We thank the Referee for this suggestion. In response to the Referee’s comments, we revised Figs 1C, S1A, S3B, and S3C so that the expression from the transgenes or the endogenous loci is more clearly labelled, the expression levels are shown in a heatmap representation, and redundant information is removed. We believe that the revised figures better convey key information and we sincerely hope the Referee’s approval.

Figure 1 - panel F - is there a reason why significance is not calculated in these graphs?

Response 4. The key message of this panel is that the Dox-induced expression of any combination of *SOX17*, *TFAP2C* and *BLIMP1* in iMeLCs does not lead to the induction of *BLIMP1*-tdTomato (BT)⁺ and *TFAP2C*-EGFP (AG)⁺ hPGCLCs, which we think should be clear from the data. We assume that the abbreviations we used and the fact there was essentially no induction in the Dox-induced columns made the key message difficult to grasp; therefore, we revised the panel with respect to these points. We also indicated the numbers of the experiments we performed in the revised manuscript (Fig 1F and its legend in the revised manuscript).

Figure 2 - the colour choice is poor as it does not allow the appreciation of the data with the different experimental conditions and it is not easy to distinguish the 'pale' colours that the authors describe - please improve this.

Response 5. We thank the Referee for pointing this out. To enhance the visibility of the data plots for the control induction, we used larger, clearly delineated symbols and directly annotated the plots (Fig 2A). Since the key message of Figs 2A-2E is that the Dox-induced expression of any combination of *SOX17*, *TFAP2C* and *BLIMP1* in iMeLCs does not lead to the induction of hPGCLCs, we removed the plots representing the induction by BMP4 and Dox from Fig 2B and 2C. Since the data under different experimental conditions (different transcription-factor inductions) are presented in separate panels (Figs 2A-2E), we think they should be appreciated easily. We sincerely hope that the revised Figure meets with the Referee’s approval.

Figure 2 - panel A - the 'developmental trajectory' is said to be highlighted in gray - how was this calculated? Is this pseudotiming? Or velocity based? - please provide the computational details of the analysis.

Response 6. As described in the main text and figure legend, the data shown in Figs 2A-2E are principal component analyses (PCA), in which the degrees of the differences between the samples are projected in a linear manner on the PCA plain, and therefore, one can evaluate the differentiation progression by simply tracing the differentiation time points, which are indicated by the grey arrow. We provided a succinct explanation in the legend to Fig 2A in the revised manuscript.

Figure 5H - the GATA2/3 WT cells are very dispersed - and it would seem that the outlier with the highest value may 'bias' the analysis - can the authors add an additional replicate here?

Response 7. As reported in our previous manuscripts, the efficiency for hPGCLC induction from hiPSCs varies to this extent (~20%–60%), although the underlying reason remains unclear (4-6). We therefore believe that the data for the efficiency for hPGCLC induction from parental hiPSCs, which was performed side-by-side with hPGCLC induction from the knockout lines, represents a variation within a normal range. In response to the Referee's comment, we provided an explanation on this point in the revised manuscript (the legend to Fig 5F in the revised manuscript).

Figure 6 is missing n numbers throughout panels A, B and C. How reproducible was this? How many times was this analyses performed with the same results? How many cells analysed?

Response 8. We thank the Referee for pointing this out. We provided a panel showing the numbers of embryos, total sections, and sections with PGCs, PGCs, and GATA3⁺ PGCs in Fig 6D in the revised manuscript.

Figure 7 panel B - n numbers and reproducibility information is missing

Response 9. We thank the Referee for pointing this out. We performed two independent experiments, analyzed 7 sections 100 μm apart in both experiments, and counted the numbers of hPGCLC-derived cells (marked by human mitochondria antibody staining) and TFAP2C/DDX4-expressing cells. From the first experiment, we detected 28 TFAP2C/DDX4-expressing cells, and from the second experiment, 23 TFAP2C/DDX4-expressing cells. We provided key information in the legend to Fig 7B in the revised manuscript.

Figure 7 - panel D - what are the red points on the graphs?

Response 10. We thank the Referee for pointing out this error. The red points show

the expression levels of the indicated genes in cells derived from the *GATA3/SOX17/TFAP2C* clone at the designated time points. We provided an explanation in the legend to Fig 7D in the revised manuscript.

Figure EV3 - n numbers are missing in panels D and E.

Response 11. We performed the transgene-induction experiments 6 and 2 times for the *GATA3/SOX17/TFAP2C* and *GATA3/SOX17* clone, respectively. The figures show representative images. We provided the n numbers in the legend to Fig S3D-E in the revised manuscript.

Page 5 - the conclusion on a progressive maturation of hPGCLC is a little inadequate, since the authors only looked at two timepoints.

Response 12. Please note that the first paragraph of the **RESULTS** section describes a method for hPGCLC induction that we have reported previously, in which we analyzed the hPGCLC differentiation process every two days up to induction day 10 and showed that hPGCLCs mature progressively during this period (4-6). In the revised manuscript, we presented this paragraph in the present tense in order to make it clear that this paragraph describes a previously established method for hPGCLC induction (the “*SOX17, TFAP2C, and BLIMP1 are not sufficient to generate hPGCLCs*” section, the first paragraph in the revised manuscript).

Page 7 - 'they lacked sufficient expression of genes specifying hPGCLC propoerties' - what does this mean? What expression levels are 'sufficient', compared to what?

Response 13. We are sorry for the insufficient explanation of the data shown in Fig EV2A. As described in the original manuscript (and in the revised manuscript as well), we previously identified 481 genes that are up- or down-regulated during the hiPSC-to-hPGCLC differentiation (6), and these are shown in the PCA plot in Fig EV2A.

The genes colored in red (1st quadrant) are defined as the PGCLC-specific genes, the genes in yellow (2nd and 3rd quadrant) are the pluripotency genes, and the genes in blue (4th quadrant) are mesoderm/endoderm-associated genes. The colors in the sidebar of the heatmap shown in Fig EV2A correspond to these genes. As shown in the heatmap, the 1st quadrant genes (red) are expressed at lower levels in the transcription factors-induced cells (right columns) compared to BMP4-induced day 2 or day 4 hPGCLCs. We provided these explanations in the revised manuscript (the legend to Fig S2A in the revised manuscript).

Page 6-7 - the part on the TF description and their potential relevance could benefit from ATACseq data (e.g. analysis of footprints, rather than expression based on transcriptomes) - if the authors have such data available. If this is not available, I

would recommend to fully removed or drastically trim this section: it's only really descriptive and too long.

Response 14. We would like to perform a detailed analysis of the mechanisms of actions of the relevant transcription factors in a future study.

In response to the Referee's comment, we trimmed this segment in the revised manuscript (the "**SOX17, TFAP2C, and BLIMP1 are not sufficient to generate hPGCLCs**" section in the revised manuscript).

Page 11 - conclusion on SOX17 and TFAP2C driving the hPGCLC specification programme is a little overstated- in reality the conclusions there are restricted to the 'transcriptional' programme –

Response 15. In response to the Referee's comment, we revised the sentence accordingly (the "**GATA TFs, SOX17, and TFAP2C drive hPGCLC induction**" section, the eighth paragraph in the revised manuscript).

Page 13, the primate embryo section, while interesting, is extremely long and rambles a bit - I suggest to streamline, since it is anyway only descriptive. Instead, the authors could use this extra 'space' to describe a bit better the WGBS analysis on G3SA-derived cells.

Response 16. In response to the Referee's comment, we streamlined the "**GATA3 expression in post-implantation primate embryos**" section in the revised manuscript.

We provided a detailed description of the results of the whole genome bisulfite sequence analysis of oogonia/gonocyte-like cells induced from hPGCLCs in our previous manuscripts (7, 8). Since the methylation profiles of the *GATA3/SOX17/TFAP2C* clone-derived cells shown in Figs 7E-7H in the original/revised manuscript are very similar to those described previously, we left the section as it was and provided an additional recent reference (8) in the revised manuscript (the "**TF-induced hPGCLCs are competent for epigenetic reprogramming and differentiation into oogonia/gonocytes**" section, the third paragraph in the revised manuscript).

REFERENCES

1. G. Martins, K. Calame, Regulation and functions of Blimp-1 in T and B lymphocytes. *Annu Rev Immunol* **26**, 133-169 (2008).
2. E. K. Bikoff, M. A. Morgan, E. J. Robertson, An expanding job description for Blimp-1/PRDM1. *Curr Opin Genet Dev* **19**, 379-385 (2009).
3. S. Nagaoka, I. *et al.*, ZGLP1 is a determinant for the oogenic fate in mice.

- Science* **367**, (2020).
4. K. Sasaki *et al.*, Robust In Vitro Induction of Human Germ Cell Fate from Pluripotent Stem Cells. *Cell Stem Cell* **17**, 178-194 (2015).
 5. S. Yokobayashi *et al.*, Clonal variation of human induced pluripotent stem cells for induction into the germ cell fate. *Biol Reprod* **96**, 1154-1166 (2017).
 6. Y. Kojima *et al.*, Evolutionarily Distinctive Transcriptional and Signaling Programs Drive Human Germ Cell Lineage Specification from Pluripotent Stem Cells. *Cell Stem Cell* **21**, 517-532 e515 (2017).
 7. C. Yamashiro *et al.*, Generation of human oogonia from induced pluripotent stem cells in vitro. *Science* **362**, 356-360 (2018).
 8. Y. Murase *et al.*, Long-term expansion with germline potential of human primordial germ cell-like cells in vitro. *EMBO J in press*, (2020).

January 6, 2021

RE: Life Science Alliance Manuscript #LSA-2020-00974-TR

Prof. Mitinori Saitou
Institute for the Advanced Study of Human Biology, Kyoto University
Yoshida-Konoe-cho, Sakyo-ku
Kyoto 606-8501
Japan

Dear Dr. Saitou,

Thank you for submitting your revised manuscript entitled "GATA transcription factors, SOX17 and TFAP2C drive the human germ-cell specification program". We would be happy to publish your paper in Life Science Alliance pending final revisions necessary to meet our formatting guidelines.

Along with the points listed below, please also attend to the following,

- please upload your main and supplementary figures as single files
- please check the legend for Figure 4 (there is mentioned panel I but in the actual figure, there is panel H)
- please add a callout for Figure 1C to your main manuscript text
- some of the graphs are repeated between Figure 2H and Figure S2B. We request you to clarify this point in the manuscript text and the figure legend

A. FINAL FILES:

-- Summary blurb (enter in submission system): A short text summarizing in a single sentence the study (max. 200 characters including spaces). This text is used in conjunction with the titles of

papers, hence should be informative and complementary to the title. It should describe the context and significance of the findings for a general readership; it should be written in the present tense and refer to the work in the third person. Author names should not be mentioned.

B. MANUSCRIPT ORGANIZATION AND FORMATTING:

Sincerely,

Shachi Bhatt, Ph.D.
Executive Editor
Life Science Alliance
<https://www.lsjournal.org/>
Tweet @SciBhatt @LSAJournal

February 5, 2021

RE: Life Science Alliance Manuscript #LSA-2020-00974-TRR

Prof. Mitinori Saitou
Institute for the Advanced Study of Human Biology, Kyoto University
Yoshida-Konoe-cho, Sakyo-ku
Kyoto 606-8501
Japan

Dear Dr. Saitou,

Thank you for submitting your Research Article entitled "GATA transcription factors, SOX17 and TFAP2C drive the human germ-cell specification program". It is a pleasure to let you know that your manuscript is now accepted for publication in Life Science Alliance. Congratulations on this interesting work.

DISTRIBUTION OF MATERIALS:

Again, congratulations on a very nice paper. I hope you found the review process to be constructive and are pleased with how the manuscript was handled editorially. We look forward to future exciting submissions from your lab.

Sincerely,

Shachi Bhatt, Ph.D.

Executive Editor

Life Science Alliance

<https://www.lsjournal.org/>

Interested in an editorial career? EMBO Solutions is hiring a Scientific Editor to join the international Life Science Alliance team. Find out more here -

https://www.embo.org/documents/jobs/Vacancy_Notice_Scientific_editor_LSA.pdf